# Molecular profiling of aromatase inhibitor sensitive and resistant ER+HER2-postmenopausal breast cancers

Eugene F. Schuster [1,2] ✉, Elena Lopez-Knowles[1,2], Anastasia Alataki[1,2], Lila Zabaglo[3], Elizabeth Folkerd[1,2], David Evans[3], Kally Sidhu[3], Maggie Chon U. Cheang[4], Holly Tovey[4], Manuel Salto-Tellez[5,6,7], Perry Maxwell[5], John Robertson[8], Ian Smith[9], Judith M. Bliss[4] & Mitch Dowsett [9]

Aromatase inhibitors (AIs) reduce recurrences and mortality in post-menopausal patients with oestrogen receptor positive (ER+) breast cancer (BC), but >20% of patients will eventually relapse. Given the limited understanding of intrinsic resistance in these tumours, here we conduct a large-scale molecular analysis to identify features that impact on the response of ER + HER2- BC to AI. We compare the 15% of poorest responders (PRs, *n* = 177) as measured by proportional Ki67 changes after 2 weeks of neoadjuvant AI to good responders (GRs, *n* = 190) selected from the top 50% responders in the POETIC trial and matched for baseline Ki67 categories. In this work, low *ESR1* levels are associated with poor response, high proliferation, high expression of growth factor pathways and non-luminal subtypes. PRs having high *ESR1* expression have similar proportions of luminal subtypes to GRs but lower plasma estradiol levels, lower expression of estrogen response genes, higher levels of tumor infiltrating lymphocytes and immune markers, and more *TP53* mutations.

The lifetime risk of a woman presenting with breast cancer (BC) is now 1 in 7[1]. Over 80% of cases are oestrogen receptor positive (ER+) and that proportion is expected to increase over the coming years. Virtually all patients with primary ER + BC receive at least 5 years of endocrine therapy targeted at reducing or eliminating oestrogenic signalling. In postmenopausal women, this is most frequently with an aromatase inhibitor (AI) because these drugs have been shown to reduce mortality from BC by approximately 40%; nevertheless, reducing this further is a priority for contemporary research, as patients with ER + BC have a steady risk of recurrence for 15 years or more after treatment. More than 20% of patients with ER+ tumours will eventually

have a relapse[2]; therefore, there is an urgent need to understand the mechanisms that underpin intrinsic and acquired resistance to oestrogen deprivation therapy[3].

ER + BC has been well studied with many pathways and genomic alterations being characterized in untreated patients and associated with recurrence rates, but there have been few large-scale studies looking at response to oestrogen depletion. Importantly, there are several gene expression profile assays that have emerged from large-scale datasets of primary BC that can accurately estimate the individual risk of recurrence but do not predict response to treatment[4–6]. Only the Breast Cancer Index (BCI) HOXB13/IL17BR ratio (H/I) has been

[1]The Breast Cancer Now Toby Robins Research Centre at the Institute of Cancer Research, London, UK. [2]Ralph Lauren Centre for Breast Cancer Research, Royal Marsden Hospital, London, UK. [3]UK NEQAS ICC & ISH, London, UK. [4]Clinical Trials and Statistics Unit, Division of Clinical Studies, The Institute of Cancer Research, London, UK. [5]Precision Medicine Centre of Excellence, The Patrick G Johnston Centre for Cancer Research, Queen's University Belfast, Belfast, UK. [6]Cellular Pathology, Belfast Health and Social Care Trust, Belfast City Hospital, Belfast, UK. [7]Division of Molecular Pathology, The Institute of Cancer Research, London, UK. [8]Faculty of Medicine & Health Sciences, Queen's Medical Centre, Nottingham, UK. [9]Royal Marsden Hospital, London, UK. ✉e-mail: gene.schuster@icr.ac.uk

shown to predict of benefit from endocrine therapy[7]. In addition, the difficulty of generating ER+ cell lines from patients has limited pre-clinical studies in AI resistance, and most of the pre-clinical studies that have been conducted to date are associated with acquired AI resistance, which can be very different from intrinsic resistance[8].

Studies investigating the association of biomarkers with treatment resistance that depend on disease recurrence as their index of resistance are confounded by prognostic clinical factors, such as tumour size and nodal status. Surgery can eradicate disease in some patients and thereby remove the potential for recurrence but there are currently no tools to distinguish such patients from those that do not recur as a result of endocrine treatment alone. Studying the biologic response to AI in the presurgical setting is therefore an attractive option that we and others have exploited over recent years to understand intrinsic resistance to AI. While clinical response, i.e. shrinkage of the tumour, can be used as the marker of response, most of these studies have used change in the proliferative marker Ki67 as the index of responsiveness[9-11] since this has been found to predict benefit from adjuvant endocrine therapy.

In the current study, we drew on biopsy samples taken from nearly 3000 postmenopausal patients with ER + BC treated presurgically with 2 weeks of an AI from the PeriOperative Endocrine Treatment for Individualised Care (POETIC) clinical trial. Ki67 was measured at diagnosis and at surgery in the tumours of these patients and it was confirmed that Ki67 after 2-weeks' treatment is more prognostic of 5-year recurrence risk than baseline Ki67 in ER + HER2- BC: recurrences were 60% lower for patients with baseline Ki67 ≥10% that fell below 10% after 2 weeks of AI (8.4% recurrence within 5 years) compared to patients whose Ki67 remained high (≥10%) after AI (21.7% recurrence within 5 years)[12], suggesting intrinsic AI resistance is a major factor in 'early' recurrence (< 5 years). Additional follow-up work will be undertaken to determine if the relationship of Ki67 response to AI with later recurrences persists beyond 5 years after diagnosis and to determine associations between clinicopathological/molecular features and recurrence despite good response to AI.

To identify correlates of AI response, we undertook analysis of a diverse set of biomarkers, including comprehensive transcriptome RNA sequencing for discovery, plasma estradiol levels due to our previous report of the strong association between estradiol levels and expression of oestrogen-responsive genes[13], immune markers and tumour-infiltrating lymphocytes (TILs) due to our earlier report of an association between immune-related gene expression and AI resistance[14], and targeted DNA sequencing of genes known to be frequently mutated in BC[15,16] or have been associated with AI resistance[8]. Patients with the poorest anti-proliferative response to AI were matched to patients with good responses. Importantly, matching was based on baseline Ki67 proliferation categories to ensure that the good (GRs) and poor responders (PRs) had a similar risk of recurrence based on at least that factor. This strategy aimed to avoid an analysis model in which good responders would be enriched for low proliferation and therefore better prognosis tumours.

## Results

### Patient cohort
The design and primary results of the POETIC trial are detailed elsewhere[12]. In brief, 4480 postmenopausal women with primary ER + BC were randomised 2:1 to receive either treatment with a non-steroidal AI (letrozole or anastrozole) for 2 weeks before and 2 weeks after surgery or to no perisurgical treatment. Only AI-treated patients with HER2- tumours, paired baseline and surgery Ki67 available, and baseline Ki67 immunohistochemistry (IHC) > 10% (to minimise imprecision in proportional Ki67 falls) were included for selection. The 15% of patients in the AI-treated group that showed the least proportional fall in Ki67 (PRs) were selected and matched to GRs from the 50% of patients showing the greatest proportional fall in Ki67. As the

average baseline Ki67 IHC for PRs was more than 50% higher than the average of all baseline Ki67 IHC in POETIC, GRs were matched to PRs based on baseline Ki67 categories (10–20%, 20–30%, and ≥30%) to ensure similar baseline proliferation rates.

In the POETIC trial, 67% of the ER + HER2- BC had Ki67 IHC > 10% and were therefore eligible for this study. Thus, the included PRs and GRs represented 10% and 34% of the total ER + HER2- population in POETIC, respectively (Supplementary Fig. 1a). In total, 367 (190 GRs; 177 PRs) and 341 (174 GRs; 167 PRs) paired sample sets from AI-treated patients were available for RNA-seq and targeted exome sequencing, respectively. A few unpaired samples were sequenced but not included in the analysis. A consort diagram (Supplementary Fig. 1b) shows the reasons for sample availability. The demographics for these patients are shown in Table 1. In general, the GRs and PRs had similar clinicopathological characteristics except for greater number of PRs that were <59 years old, had grade 3 tumours, or had chemotherapy.

### Oestrogen receptor levels
Expression of ER is an established determinant of responsiveness to endocrine therapies. It is well known that there is a good relationship between *ESR1* and ER protein expression in BC[17] and we have reported this separately for the POETIC trial[18]. Our first analysis was therefore to assess the relationship between *ESR1* expression from the RNA-seq data. Figure 1a shows that only 2 of the 190 GRs had *ESR1* levels below 12 log2 normalised counts compared with 58 out of the 177 PRs. Thus, low *ESR1* amongst these cases that were diagnosed locally as ER+ appears to be a major determinant of poor response in approximately a third of patients. We therefore created 2 categories of PRs with values of ESR1 above and below 12 for log2 normalised counts (PRs ESR1[HIGH] and PRs ESR1[LOW], respectively). Hierarchical unsupervised clustering of genes differentially expressed between GRs and all PRs is shown in Fig. 1b and emphasizes the major difference between PRs ESR1[HIGH] and PRs ESR1[LOW] tumours with the latter segregating almost completely from all other tumours (to the right of Fig. 1b). This is also illustrated by the greater than 10-fold difference of the baseline expression of the ER-regulated genes *TFF1* and *PGR* between PRs ESR1[HIGH] and PRs ESR1[LOW] (FDR < 10^−10 for both genes in DESeq2 analysis) (Fig. 1d, f; Supplementary Data 1). PgR IHC and *PGR* RNA-seq values were highly correlated for baseline samples (Spearman rho = 0.87, $p < 10^{-100}$) (Supplementary Fig. 2a), and PgR protein expression was also significantly higher in GRs compared with PRs ESR1[HIGH] and PRs ESR1[LOW] ($p = 1 \times 10^{-5}$ and $3 \times 10^{-23}$, respectively; Mann–Whitney test) (Fig. 1h). PgR was not detected using IHC in 2% (4/183) of GRs, 12% (13/111) of PRs ESR1[HIGH] and 77% (43/56) of PRs ESR1[LOW] at baseline. *MKI67* gene expression and Ki67 IHC were significantly higher in PRs ESR1[LOW] compared with PRs ESR1[HIGH] and GRs (FDR = 0.003, DESeq2 analysis and $p = 5.5 \times 10^{-7}$, Mann–Whitney analysis, respectively) but not PRs ESR1[HIGH] compared to GRs (Fig. 1e, I; Supplementary Data 1). PRs ESR1[LOW] are associated with a significantly higher percentage of grade 3 tumours compared to GRs ($p = 0.0003$, Fisher-exact) and PRs ESR1[HIGH] ($p = 0.03$, Fisher-exact), and grade 3 tumours are much more likely to be offered chemotherapy.

In addition, principal component analysis (PCA) shows clear separation between PRs ESR1[LOW] and tumours expressing high *ESR1* (GRs and PRs ESR1[HIGH]) (Fig. 1j). Only 15% of the genes expressed significantly differently between all PRs and GRs are also significantly differentially expressed between ESR1[HIGH] and GRs, while 75% are significantly differentially expressed between PRs ESR1[LOW] and PRs ESR1[HIGH], also suggesting that the differences between ESR1[LOW] and ESR1[HIGH] samples dominate any comparison (Supplementary Fig. 2c, Supplementary Data 1). The top two differentially expressed genes between GRs and both all PRs and PRs ESR1[LOW] were *ESR1* (down in PRs) and *FOXC1* (up in PRs), a gene known to be associated with basal-like BC[19]; neither of them are significantly different between GRs and PRs ESR1[HIGH] (Supplementary Data 1). Further comparisons were therefore

**Table 1 | The demographics of patients in study separated by GRs and PRs**

| | GRs $n = 190$ | PRs $n = 177$ | |
|---|---|---|---|
| **Age at randomisation (years)** | | | |
| Median | 68 | 66 | |
| <59 | 19% (37) | 29% (51) | $p = 0.003$ (Fisher-exact) |
| 60–69 | 39% (75) | 34% (61) | |
| 70–79 | 29% (56) | 24% (42) | |
| ≥80 | 11% (21) | 12% (22) | |
| Unknown | 1% (1) | 1% (1) | |
| **Tumour size (cm)** | | | |
| Median | 2 | 2.1 | |
| ≤2 | 55% (104) | 46% (81) | |
| >2 & ≤5 | 43% (82) | 51% (91) | |
| >5 | 2% (3) | 1% (2) | |
| Unknown | 1% (1) | 1% (2) | |
| **Tumour grade** | | | |
| G1 | 8% (16) | 7% (12) | |
| G2 | 61% (115) | 45% (79) | $p = 0.002$ (Fisher-exact) |
| G3 | 24% (45) | 41% (72) | $p = 0.001$ (Fisher-exact) |
| Unknown | 7% (14) | 8% (14) | |
| **Nodal status** | | | |
| 0 | 59% (113) | 56% (100) | |
| 1–3 | 28% (53) | 31% (55) | |
| ≥3 | 12% (23) | 12%(12%) | |
| **Chemotherapy** | | | |
| Yes | 29% (43) | 42% (75) | $p = 0.0001$ (Fisher-exact) |
| Unknown | 2% (3) | 1% (2) | |
| **Vascular invasion** | | | |
| No | 61% (115) | 58% (103) | |
| Yes | 35% (67) | 34% (60) | |
| Unknown | 4% (8) | 8% (14) | |
| **Histological type** | | | |
| IDC | 83% (157) | 88% (155) | |
| ILC | 13% (25) | 10% (17) | |
| Other | 2% (4) | 2% (4) | |
| Unknown | 2% (4) | 1% (1) | |
| **Ki67 % Baseline** | | | |
| Median | 26% (range 10% to 73%) | 29% (range 10% to 97%) | |
| **Ki67 % 2wk** | | | |
| Median | 2% (range 0% to 11%) | 24% (range 6% to 95%) | $p = 1.16 \times 10^{-60}$ (Mann–Whitney) |
| **Ki67 % Change** | | | |
| Median | −91% (range −80 to −100%) | −14% (range 48% to 184%) | $p = 1.42 \times 10^{-61}$ (Mann–Whitney) |

Significant differences ($p < 0.05$) between Grs and PRs determined by two-sided Fisher's exact test for categories and by two-sided Mann–Whitney test for continuous variables (Ki67).

focused largely on identifying differences between GRs and PRs ESR1[HIGH] although we also present the PRs ESR1[LOW] for completeness.

**Changes in gene expression with AI**

Suppression of oestrogen response (as represented by *PGR*) between baseline and 2 weeks for PRs ESR1[HIGH] was observed to be similar to GRs (log2 FC 1.5 and 1.8, respectively) (Fig. 1k). Differences in proliferation were not similar (as represented by *MKI67* with log2 FC of −0.4 and −2.3, and % change in Ki67 of −12% and −91% in PRs ESR1[HIGH] and GRs, respectively) (Fig. 1l, m). These data indicate that PRs ESR1[HIGH] had a

response to AI that was distinct from both GRs and PRs ESR1[LOW] and importantly, this suggests a disconnect between signalling that controls classical oestrogen-responsive genes and that controlling proliferation in PRs ESR1[HIGH].

**Intrinsic subtyping**

Intrinsic subtyping was determined by adapting the gene-level median centering approach[20] (see 'Methods', Supplementary Fig 2d, e). PRs ESR1[LOW] were highly enriched with non-luminal subtypes (62% Basal, 29% HER2-enriched [HER2-e]), and 20% (1/5) luminal subtypes in this group were high confidence calls (>0.95) compared to 92% (49/53) non-luminal high confidence calls. The GRs and PRs ESR1[HIGH] were similar to one another in their enrichment of luminal subtypes (Fig. 2a, Supplementary Fig. 3a–d): GRs included 1% Basal, 1% HER2-e, 42% Luminal A (LumA), 56% Luminal B (LumB) cases, while PRs ESR1[HIGH] included 0% Basal, 8% HER2-e, 35% LumA, 53% LumB cases. Thus, other than a significantly higher proportion of HER2-enriched cases in the PRs ESR1[HIGH] than in GRs ($p = 0.01$, Fisher-exact), there was little difference in intrinsic subtypes between these two response groups. Figure 2b shows the individual cases associated with the different subtypes according to *ESR1* expression and % change in Ki67. Of particular note, there was no distinct separation between LumA and LumB tumours among either the GRs or PRs ESR1[HIGH] based on the change in Ki67. Figure 2c shows hierarchical supervised clustering of PAM50 gene expression separately for each of the response subgroups with normalization across the whole population. It emphasizes the major difference between the PRs ESR1[LOW] and the other two groups and the relatively modest difference between the GR from the PRs ESR1[HIGH]. However, two genes, *EGFR* and *FGFR4*, the two growth factor receptor genes in the PAM50 set, showed a significant difference (FDR = 0.0006 and FDR = 0.0002, respectively, DESeq2; Supplementary Data 1) between GRs and PRs ESR1[HIGH] with both these genes being more highly expressed in PRs ESR1[HIGH]. *FGFR4* has been associated with HER2-e subtypes and AI endocrine therapy resistance[21–23] and is significantly differentially expressed between GRs and PRs ESR1[HIGH] (FDR = 0.0002, DESeq2; Supplementary Data 1). Both *FGFR4* and *CLCA2* (FDR = 0.0002 GRs vs. PRs ESR1[HIGH]) showed high expression specific to HER2-e subtypes but not the *HER2/ERBB2* gene (Supplementary Fig. 3e-g, Supplementary Data 1). Similarly, *FOXC1* expression was specific to Basal subtypes (Supplementary Fig. 3h).

**Breast Cancer Index (BCI) *HOXB13/IL17BR* ratio (H/I)**

The *HOXB13/IL17BR* ratio (H/I) has been shown to be predictive of benefit from endocrine therapy and extended endocrine treatment with low scores showing significant benefit[7]. We observed the mean baseline H/I from the RNA-seq data was higher in PRs compared to GRs (mean H/I GRs = 0.32, PRs ESR1[HIGH] = 0.39, and PRs ESR1[LOW] = 0.49) with H/I significantly higher in PRs ESR1[HIGH] and PRs ESR1[LOW] compared to GRs ($p = 0.048$ and $p = 0.0006$, respectively; Mann–Whitney), consistent with evidence that H/I can predict benefit from endocrine therapy.

**Annotation enrichment**

A large number of genes were significantly (FDR < 0.05) differentially expressed at baseline (2034 of 16,832 expressed genes) (Supplementary Data 1). Genes expressed significantly higher in PRs ESR1[HIGH] compared to GRs were enriched for immune-related gene sets (e.g. adaptive immune response, T-cell activation, allograft rejection, and interferon-gamma response), while genes lower in PRs ESR1[HIGH] were enriched for oestrogen response genes (Supplementary Data 2).

**Gene set enrichment analysis (GSEA)**

GSEA revealed 17 hallmarks that were significantly different between GRs and PRs ESR1[HIGH] with only two being expressed to a lesser extent in PRs ESR1[HIGH]: the oestrogen response early and late gene sets

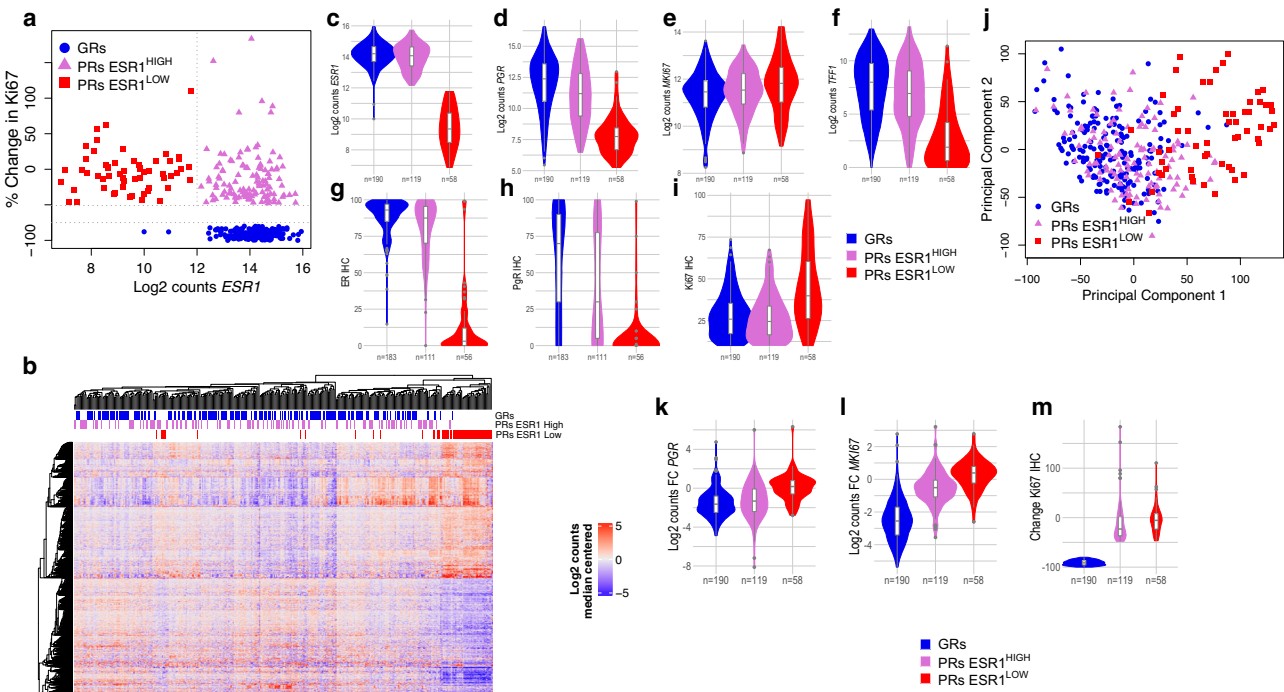

**Fig. 1 | Gene and protein expression in GRs and PRs. a** Change in Ki67 scores after 2wks AI vs. log2 ESR1 normalised counts from RNA-seq data (red = PRs ESR1[LOW]; orchid = PRs ESR1[HIGH]; blue = GRs). **b** Unsupervised heatmap of median centred log2 gene expression values from median of genes significantly (FDR < 0.05) differentially expressed and log2 fold change (FC) > 2 between GRs and PRs. Violin/Boxplot of log2 normalised counts for *ESR1* (**c**), *PGR* (**d**), *MKI67* (**e**), and *TFFI* (**f**) genes and baseline ER (**g**), PgR (**h**) and Ki67 IHC (**i**) for GRs, PRs ESR1[LOW] and PRs ESR1[HIGH]. **j** PCA plot based on expression of all genes with GRs, PRs ESR1[LOW], and PRs ESR1[HIGH]. Violin/Boxplot of change in *PGR* (**k**), *MKI67* (**l**), and Ki67 IHC (**m**) after two weeks of AI for GRs, PRs ESR1[LOW] and PRs ESR1[HIGH]. Boxplots present 25th, 50th (median), and 75th percentile values. Whiskers extend no larger than ±1.5 times the inter-quartile range with outliers plotted individually beyond this range. The number of independent samples used for comparisons between GRs and PRs is shown. Source data are provided as a Source data file.

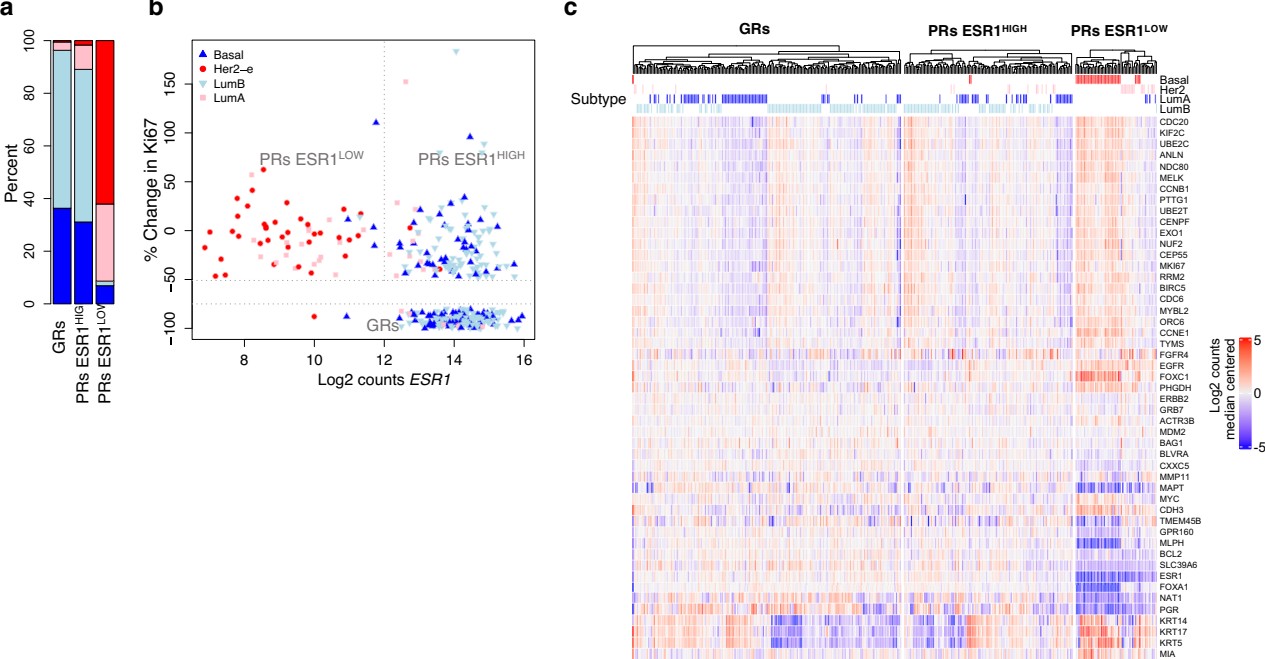

**Fig. 2 | Intrinsic suptyping of GRs and PRs. a** Percent of PAM50 subtype for GRs, PRs ESR1[HIGH] and PRs ESR1[LOW]. **b** Change in Ki67 scores after 2wks AI vs. log2 ESR1 normalised counts from RNA-seq data marked by PAM50 subtype (LumA = blue; LumB = light blue; HER2-enriched=pink; Basal = red). **c** Heatmaps of median-centred log2 gene expression values supervised by GRs, PRs ESR1[LOW] and PRs ESR1[HIGH] of PAM50-subtyping genes (log2 FC from median) for POETIC data. Labelled with PAM50 subtype (LumA = blue; LumB = light blue; HER2-enriched = pink; Basal = red). Source data are provided as a Source data file.

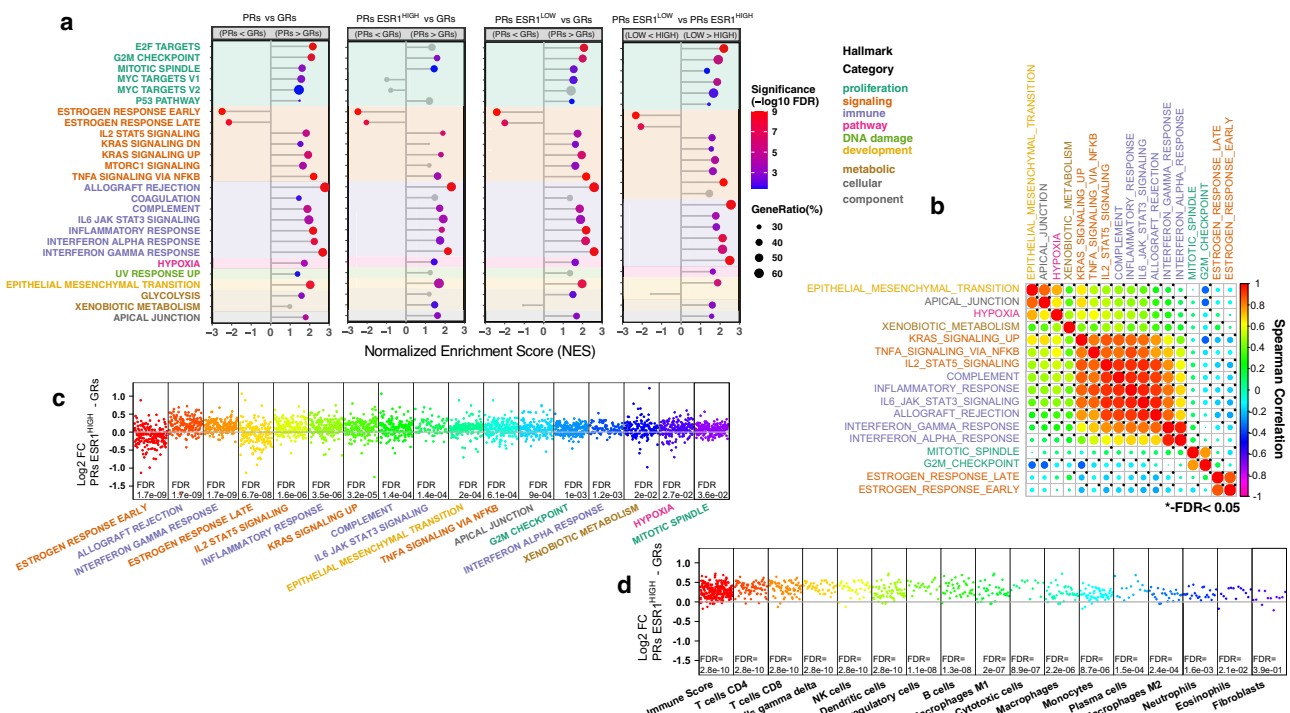

**Fig. 3 | Hallmark GSEA and TME analysis. a** Plot of normalized enrichment score (NES) of Molecular Signature Database (MSD) Hallmark gene sets from Gene Set Enrichment Analysis (GSEA) for all PRs vs GRs, GRs vs PRs ESR1$^{LOW}$ and GRs vs PRs ESR1$^{HIGH}$ and PRs ESR1$^{LOW}$ vs PRs ESR1$^{HIGH}$ comparisons. Hallmarks with FDR < 0.05 in any comparison shown. Hallmark genes sets with FDR < 0.05 are shown (red to blue colouring based on significance; grey if >0.05). Hallmarks are coloured by biological process categories as shown in legend. **b** Plot of single samples GSEA score correlations between MSD Hallmark gene sets for GRs and PRs ESR1$^{HIGH}$. Plot of log2 FC PRs ESR1$^{HIGH}$ – GRs for individual genes within MSD Hallmark gene sets that are significantly different between PRs ESR1$^{HIGH}$ and GRs in GSEA (**c**) and all Consensus Tumour Microenvironment (TME) breast cancer gene sets (**d**). FDR values from GSEA analysis. Hallmarks are coloured by biological process category. Source data are provided as a Source data file.

(Fig. 3a). Thus, oestrogen-responsive gene expression at baseline was overall lower in this PR subgroup, despite *ESR1* expression being similar to GRs.

Of particular note, about half of the hallmarks that were upregulated in PRs ESR1$^{HIGH}$ compared with the GRs were associated with immune processes, and these immune hallmarks were highly significantly correlated with each other (FDR < 0.05). In addition, the immune hallmarks were correlated with genes upregulated after KRAS activation (KRAS Signaling Up) and genes regulated by NF-κB in response to tumour necrosis factor alpha (TNFA) (TNFA Signaling via NF-κB). Both KRAS Signaling Up and TNFA Signaling via NF-κB hallmarks were also significantly higher in PRs ESR1$^{HIGH}$, and the genes within these hallmarks significantly overlapped with immune hallmarks, including interferon-gamma response (FDR = $2 \times 10^{-7}$ and FDR = $3 \times 10^{-30}$, respectively; hypergeometric distribution calculation). Similarly, hypoxia is higher in PRs ESR1$^{HIGH}$, consistent with our earlier observation of a hypoxia metagene being correlated with poor Ki67 suppression by AI[24], and positively correlated with immune hallmarks, as are two other hallmarks (apical junction and epithelial-mesenchymal transition) that showed correlation with hypoxia. Xenobiotic metabolism was higher in PRs ESR1$^{HIGH}$ and showed only modest correlation with immune-associated hallmarks. In contrast, oestrogen response hallmarks were inversely correlated with several immune-related gene sets (Fig. 3b). Finally, the two gene signatures associated with proliferation (G2M checkpoint and mitotic spindle) showed higher expression in PRs ESR1$^{HIGH}$ and were strongly correlated with one another but showed little correlation with any of the other signatures.

Correlations were also calculated between the hallmarks that were significantly different in the comparisons between any of the Ki67 response groups using data from all GRs and PRs samples. In general,

the correlations are similar to those found in Fig. 3b but with stronger negative correlations being observed between oestrogen response and immune hallmarks (Supplementary Figure 4a). The heatmap in Supplementary Fig. 4b showed the distribution of correlations and heterogeneity between patients and those patients that have high expression of interferon response genes but medium to low expression of Interleukin-2/Interleukin-6 (IL2/IL6) signalling genes.

Figure 3c showed the log2 ratio of the expression of individual genes that fall in each of the GSEA Hallmarks in PRs ESR1$^{HIGH}$ versus GRs. The large majority of genes for both interferon-gamma response (82%), interferon alpha response (75%), and TNF signalling via NF-κB (73%) were more highly expressed in the PRs ESR1$^{HIGH}$ cases. Similar patterns are observed when comparing in PRs ESR1$^{LOW}$ and GRs (Supplementary Fig. 5a).

We subjected the gene expression data to deconvolution by Consensus TME[25] analysis (Fig. 3d). The overall immune score was highly significantly greater in the PRs ESR1$^{HIGH}$ than in the GRs (*p* = 2.8e −10). All immune cell types were indicated as being significantly more highly associated with PRs ESR1$^{HIGH}$, suggesting that these tumours might be considered immune hot relative to GRs with little distinction in the immune cells involved. While some of the statistical differences for a number of the cell types were modest, this appears to be due to the number of genes associated with the cell type rather than the amplitude of the difference. Hallmark and Consensus TME analysis of PRs ESR1$^{LOW}$ versus GRs showed similar but more extreme differences (Supplementary Fig. 5, 6).

**Plasma oestrogen levels**

Given that estradiol is the predominant signal for ER-dependent tumour progression, we compared plasma estradiol levels between the three response groups. Four values above 130 pmol/l were

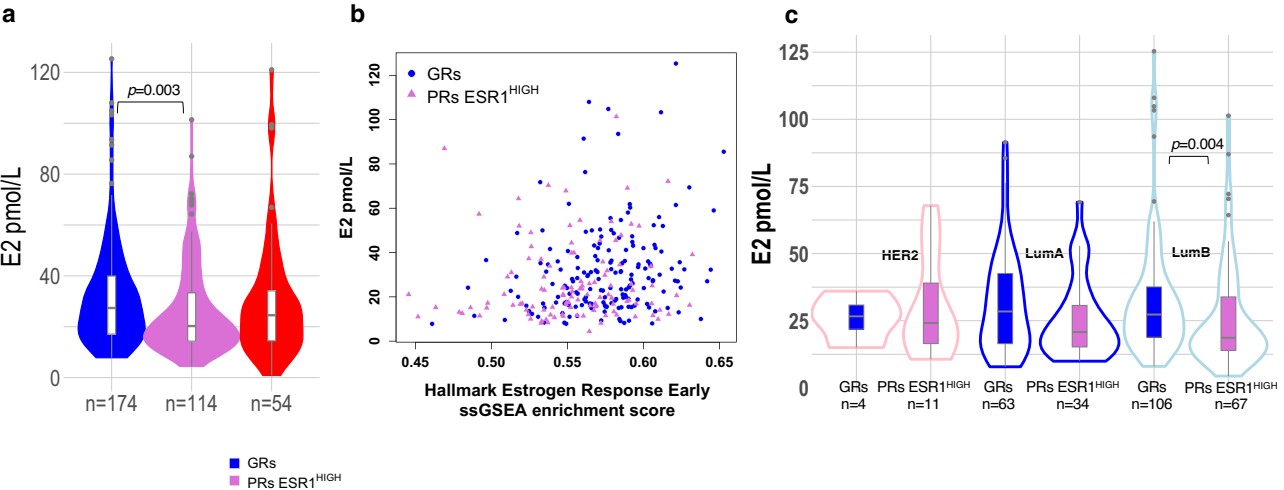

**Fig. 4 | Estradiol levels and correlations with oestrogen response. a** Violin/Boxplot of estradiol (E2) levels in POETIC patients (red = PRs ESR1$^{LOW}$; orchid = PRs ESR1$^{HIGH}$; blue = GRs). **b** Estradiol levels and single sample Gene Set Enrichment Analysis (ssGSEA) enrichment scores for Molecular Signature Database Hallmark Oestrogen Response Early genes for GRs (blue) and PRs ESR1$^{HIGH}$ (orchid). **c** Violin/Boxplot of estradiol (E2) levels in GRs (blue boxplots) and PRs ESR1$^{HIGH}$ (orchid boxplots) for HER2-E (pink outlines), LumA (blue outlines) and LumB (light blue outlines). Basal subtypes not shown due to small number of these subtypes. Boxplots present 25th, 50th (median), and 75th percentile values. Whiskers extend no larger than ±1.5 times the inter-quartile range with outliers plotted individually beyond this range. The number of independent samples used for comparisons between GRs and PRs is shown. Significant differences ($p < 0.05$) determined by two-sided Mann–Whitney tests. Source data are provided as a Source data file.

excluded from the analysis, since these were not plausibly post-menopausal. Figure 4a shows there was a significantly higher mean level of estradiol in the GRs than in the PRs ESR1$^{HIGH}$ ($p = 0.003$, Mann–Whitney test), but no difference of GRs from the PRs ESR1$^{LOW}$ in which signalling through ER would be considered minimal. However, unlike the comparison of *ESR1* expression, there was no distinct cut-off below which GRs were very unlikely to be represented. Estradiol levels were significantly correlated with expression of oestrogen response early (FDR = 0.01, Spearman) (Fig. 4b) and late (FDR = 0.02, Spearman) GSEA Hallmarks but no other hallmarks in tumours with high *ESR1* expression. Therefore, low expression of oestrogen response genes may be in part due to the effect of the lower plasma estradiol levels in the PRs ESR1$^{HIGH}$ group. In addition, two out of the top three genes that correlated with plasma estradiol levels in GRs and PRs ESR1$^{HIGH}$ are known to be regulated by oestrogen and ER (*PGR* rho = 0.26; *GREB1* rho = 0.27, Spearman) and are also expressed in significantly lower levels in PRs ESR1$^{HIGH}$ compared to GRs (Supplementary Data 2). There is a trend for lower estradiol in PRs ESR1$^{HIGH}$ compared to GRs regardless of subtype and a significant difference in LumB tumours ($p = 0.004$, Mann–Whitney) (Fig. 4c).

## Tumour-infiltrating lymphocytes (TILs)

Percentage of stromal TILs was measured for 366 out of 367 samples in this dataset. TILs were significantly higher in both PRs ESR1$^{HIGH}$ and PRs ESR1$^{LOW}$ compared to GRs (Fig. 5a), with 11% of GRs, 16% of PRs ESR1$^{HIGH}$ and 36% of PRs ESR1$^{LOW}$ being in the intermediate TILs category (11 to 59% immune cells in stroma)[26]. There were no TILs scores in the high category (>60%). As expected, TILs were highly correlated with T-cells (particularly T-regulatory cells [Tregs]), immune-related Hallmark gene sets, and PI3K/AKT/MTOR signalling, and inversely correlated with oestrogen response (Fig. 5b, c). Work by others has shown activated mTOR-dependent translation in TILs[27]. In addition, TILs were higher in PRs ESR1$^{HIGH}$ compared to GRs, regardless of subtype and significantly higher in both luminal subtypes (Fig. 5d). There was no significant correlation of TILs with estradiol levels (Supplementary Fig. 7a).

## Multiplex immunofluorescence (mIF) phenotyping

To better understand and characterize TILs, we randomly selected 25 samples from each of the GR, PRs ESR1$^{HIGH}$ and PRs ESR1$^{LOW}$ populations for mIF analysis and inspected the FFPE blocks for the amount of residual tumour. Those samples with insufficient amount of tumour were excluded and 15 samples were randomly selected from the remainder for each population.

Immune phenotyping was based on 5 markers: CD3 which is part of the T-cell receptor complex, CD20 which is a B-cell surface marker, CD68 which is transmembrane glycoprotein that is highly expressed by human monocytes and tissue macrophages, and FOXP3 which is a transcriptional regulator found in immunosuppressive Tregs. Figure 6a–f illustrates the multiplexed staining of ER, CD3, CD20, CD68, FOXP3 and CD3/FOXP3 co-localization and selection of ER-positive regions to separate tumour from stromal compartments. As expected, density of the immunophenotypic features in tumour compartments was lower than in stromal compartments, and density was the highest for CD3 and CD68. There was also a trend for a higher density of immune markers in PRs compared to GRs with significant differences in CD20 stroma density (Supplementary Fig. 7b–k)

Analysis showed a highly significant correlation between TILs and FOXP3 marker density in the stroma compartment (Fig. 6g), confirming the association of TILs with Tregs (Fig. 5b). Stromal biomarker density was highly correlated to the gene expression of the encoding gene (CD3/*CD3D*, FOXP3/*FOXP3* and CD20/*MS4A1*; *CD68* expression was too low and excluded) (Fig. 6h). There was also a strong and significant positive correlation between stromal expression of CD20, CD3, FOXP3 and CD3/FOXP3 but not CD68 with the GSEA immune hallmarks and weak negative correlation with oestrogen signalling (Fig. 6i). Similar but weaker correlations were found with immune cells expressing these markers in the tumour compartments.

The immune phenotyping using mIF showed the expected correlations with TME deconvolution with particularly strong correlations of CD20 and CD3 with the B- and T-cell gene signatures, respectively (Fig. 6j). CD68 staining correlated strongly with eosinophils and mast cell signatures as well as with the macrophage signature as expected.

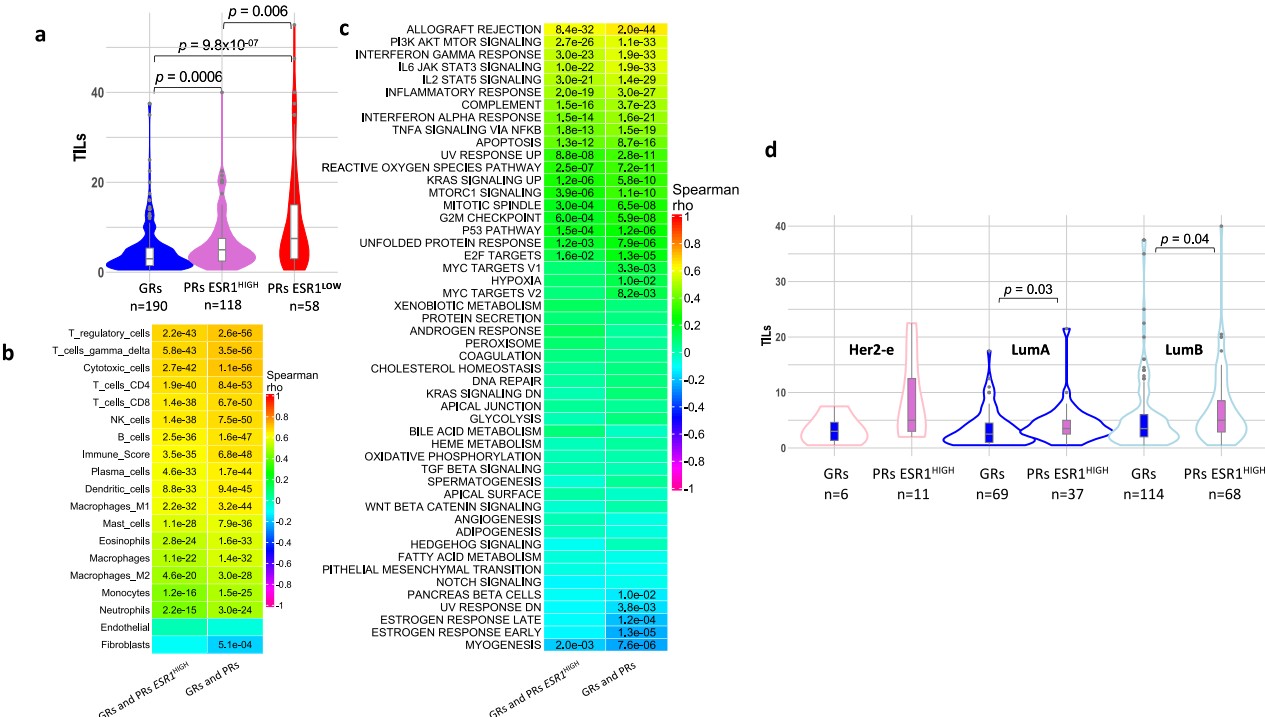

**Fig. 5 | TILs and correlations with Hallmark processes and TME. a** Violin/Boxplot of TILs per AI response group. Spearman correlations of TILs and Consensus Tumour Microenvironment (TME) breast cancer (**b**) and Molecular Signature Database Hallmark gene sets (**c**) single sample Gene Set Enrichment Analysis (ssGSEA) scores from only GRs and PRs $ESR1^{HIGH}$ or from all data GRs and PRs combined. Significant correlations (FDR < 0.05) are shown. **d** Violin/Boxplot of TILs for subtypes within GRs (blue boxplots) and PRs ESR1$^{HIGH}$ (orchid boxplots) for HER2-E (pink outlines), LumA (blue outlines) and LumB (light blue outlines). Basal subtypes not shown due to small number of these subtypes. Boxplots present 25th, 50th (median), and 75th percentile values. Whiskers extend no larger than ±1.5 times the inter-quartile range with outliers plotted individually beyond this range. The number of independent samples used for comparisons between GRs and PRs is shown. Significant differences (p < 0.05) determined by two-sided Mann–Whitney tests. Source data are provided as a Source data file.

## Somatic mutations

The proportion of patients showing mutations of the 87 sequenced were as expected for a population of ER+ tumours[16,28], with 35% being mutated for *PIK3CA*, 24% *TP53*, 10% *CDH1*, 10% *KMT2C*, 10% for *GATA3*, and 9% *MAP3K1* (Fig. 7a) with similar mutually exclusive or co-occurring mutations (Supplementary Fig. 8a). Following correction for multiple testing, none of the targeted genes showed a significant difference in mutation prevalence between the GRs and PRs ESR1$^{HIGH}$, although differences in *RB1, TP53, ARID1B* and *DNAH11* were p < 0.05 in unadjusted p-values with all showing higher prevalence in PRs ESR1$^{HIGH}$ (Fig. 7b).

There were more extreme differences in PRs ESR1$^{LOW}$ with a much higher incidence of *TP53* and *PTEN* mutations (adjusted p values < 0.05), and few if any mutations in *GATA3* or *CDH1* mutations (unadjusted p values < 0.05) (Fig. 7c). This is consistent with the basal transcriptional profile and non-luminal subtypes found in most of the tumours in that group (Supplementary Figs. 8 and 9).

There was a significant difference (p = 0.02, Mann–Whitney) between the number of mutations in GRs (mean 2.7 mutations/tumour) compared to PRs (mean 3.6 mutations/tumour), but not between GRs and PRs ESR1$^{HIGH}$ (mean 3.7 mutations/tumour) or PRs ESR1$^{LOW}$ (mean 3.3 mutations/tumour). Tumours with *TP53* mutations also had significant differences (p = 0.001, Mann–Whitney) compared to tumours with wild-type *TP53* (mean 4.2 and 2.8 mutations/tumour, respectively). *TP53* mutation was associated with higher Ki67 levels at baseline and after 2 weeks of AI (Supplementary Fig 10). There was a trend for TILs to be higher in *TP53*$^{MUT}$ tumours and significantly higher in PRs ESR1$^{HIGH}$ (Fig. 8a). For intrinsic subtypes, there was a trend for higher *TP53* mutations in PRs ESR1$^{HIGH}$ with a significant difference between *TP53*

mutations in LumA GRs and PRs ESR1$^{HIGH}$ (3% and 22%, respectively; p = 0.004, Fisher-exact; Fig. 8b).

## Copy number alterations

Copy number gains and losses occurred at similar percentages across the genome in GRs and PRs ESR1$^{HIGH}$ except for gains in PRs ESR1$^{HIGH}$ at chr6q (encoding *ESR1*, *MED23*, *FOXO3*, and *SYNE1* present in targeted exome), regions in chr2p (encoding *BIRC6* and *DNMT3A* in targeted exome), and a small region in chr9q34 although this region was not well covered by the targeted exome sequencing (Fig. 8c, Supplementary Data 3 and 4). *MED23*, *FOXO3* and *SYNE1* were expressed at significantly higher levels in PRs ESR1$^{HIGH}$ (FDR < 0.05, Supplementary Data 1).

PRs ESR1$^{LOW}$ showed many more significant differences in copy number alterations (Supplementary Fig. 11, Supplementary Data 3), especially in chromosome 16q. This is likely due to the lack of lobular tumours in PRs ESR1$^{LOW}$ (3%) compared to GRs (13%) and PRs ESR1$^{HIGH}$ (13%) and also reflected in the percent of somatic mutations in E-cadherin in each group (*CDH1* mutations 2% PRs ESR1$^{LOW}$; 13% GRs and 10% PRs ESR1$^{HIGH}$) (Supplementary Fig. 8b, c).

No significant difference in overall chromosomal instability (% of genome with copy number gain or loss) was observed between GRs and PRs ESR1$^{HIGH}$ or ESR1$^{LOW}$ and overall chromosomal instability was not correlated with TILs. However, tumours with *TP53* mutations or loss of *TP53* copy number had significantly greater percentage of the genome with copy number alterations (p = 0.0003 and p < 0.0001, respectively; Mann–Whitney, Supplementary Fig. 11b). The number of tumours with both somatic mutations and loss of one or more copy number in *TP53* were significantly higher in PRs ESR1$^{HIGH}$ (23%;

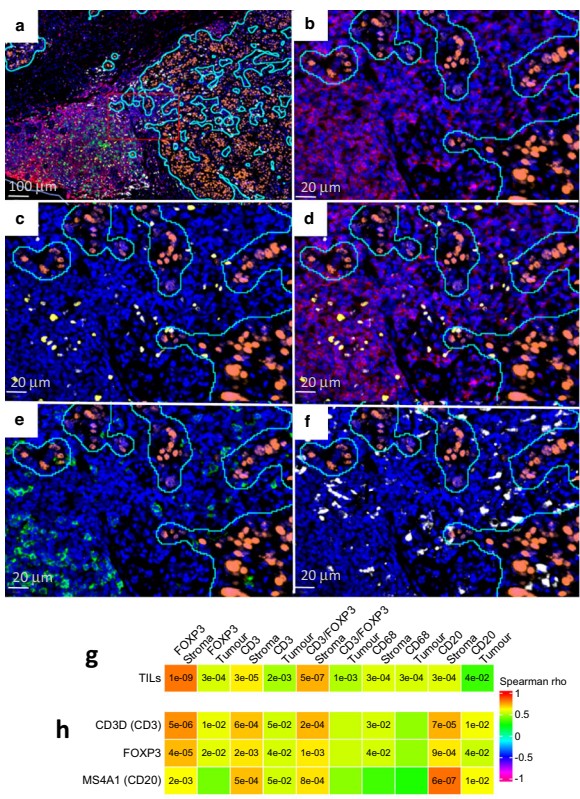

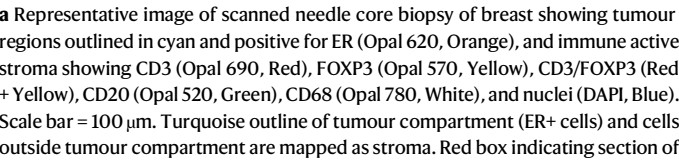

**Fig. 6 | mIF and correlations with cell markers, hallmarks and TME.**
**a** Representative image of scanned needle core biopsy of breast showing tumour regions outlined in cyan and positive for ER (Opal 620, Orange), and immune active stroma showing CD3 (Opal 690, Red), FOXP3 (Opal 570, Yellow), CD3/FOXP3 (Red + Yellow), CD20 (Opal 520, Green), CD68 (Opal 780, White), and nuclei (DAPI, Blue). Scale bar = 100 μm. Turquoise outline of tumour compartment (ER+ cells) and cells outside tumour compartment are mapped as stroma. Red box indicating section of slide shown in subsequent panels that highlight the immune active stroma: **b** CD3, **c** FOXP3, **d** CD3/FOXP3, **e** CD20, **f** CD68. Scale bar in (**b**–**f**) = 20 μm. For each

patient, a single 5 μm section was mounted on a slide, stained and cell density (cells/μm$^2$) quantified. Spearman correlations between density of immune markers for tumour/stroma compartments and expressed genes that encode the immune markers (**g**), genes encoding the protein markers except *CD68* as gene expression values were too low (**h**), single sample Gene Set Enrichment Analysis (ssGSEA) Molecular Signature Database Hallmark gene set scores (**i**), Consensus Tumour Microenvironment (TME) breast cancer gene set scores (**j**). Significant Spearman correlations (FDR < 0.05) are shown. Source data are provided as a Source data file.

$p = 0.002$, Fisher-exact) and PRs ESR1$^{LOW}$ (25%; $p = 0.008$, Fisher-exact) compared to GRs (9%) (Fig. 8d, Supplementary Fig. 11c).

## Discussion

The findings from this study confirm the wide range of pathways that are associated with intrinsic resistance to an AI, including low *ESR1*/ER expression, low expression of oestrogen response genes, HER2-e and Basal-like subtypes, higher immune-related markers, *TP53* mutations and higher growth factor expression (Fig. 9). While PRs ESR1$^{HIGH}$ and PRs ESR1$^{LOW}$ shared some AI resistance phenotypes at the pathway level (Fig. 3a), the two groups were very distinct on the molecular level (Fig. 1b) and likely to have different overall prognosis with worse outcome expected for the non-luminal subtypes[29] that are highly enriched in PRs ESR1$^{LOW}$.

While GRs clearly benefit from AI treatment, it is not known whether additional decreases in Ki67 might have occurred in PRs ESR1$^{HIGH}$ if longer AI treatment had been given, but in the IMPACT trial AI-induced suppression of Ki67 was only marginally greater after 12/14 weeks than at 2 weeks[9,30]. Some oestrogen response genes (e.g. *PGR*) are still supressed in PRs ESR1$^{HIGH}$, suggesting ER signalling is still functional in these tumours but not driving proliferation. This highlights the importance of using markers of proliferation to identify AI response instead of other surrogate markers associated with oestrogen response.

An unexpected association was lower estradiol levels in PRs ESR1$^{HIGH}$, but this observation is likely to explain the low expression of

oestrogen response genes and lack of response to AI in these patients, as proliferation is not likely to be strongly driven by oestrogen in these tumours. Higher plasma estradiol levels have been consistently reported in postmenopausal women that develop BC and this has been seen most strongly in those with ER + BC[31]. This suggests that a continuum whereby high exposure to estradiol promotes those ER tumours with the highest sensitivity and the proliferation in those tumours is the most sensitive to estradiol deprivation. We also observed gains in chr6q in PRs ESR1$^{HIGH}$, which encodes *ESR1* and two genes, *MED23*[32] and *FOXO3*[33], known to interact with *ESR1*. The transcriptional mediator MED23 plays a key role in the oestrogen-dependent BC growth and is associated with poor outcomes[32].

Beyond oestrogen-related markers, there was a strong relationship between immune markers and AI resistance, and this is concordant with our earlier observation in a different, smaller population[14]. This observation of an association of immune features with poor response to endocrine therapy contrasts with the better outcome of patients with triple negative BC seen with high TILs count[34]. In the large German Breast Group study of outcome for different subtypes according to TILs, the longer disease-free survival for triple negative disease with higher TILs was as expected but no improvement was seen in patients with ER + HER2- disease[35]. There is growing number of retrospective studies showing that immune-related biomarkers have prognostic value in ER + BC beyond higher TILs, including poor outcomes associated with higher tumour

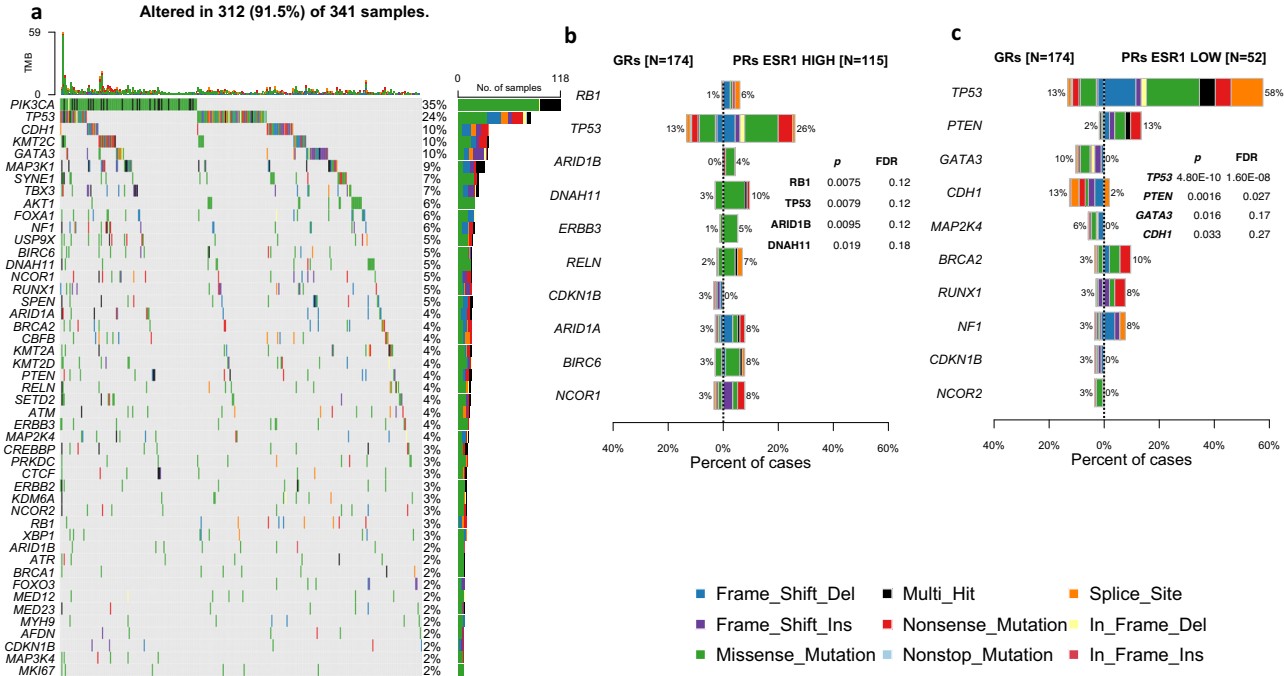

**Fig. 7 | Oncoplot and barplots of top somatic mutations. a** Oncoplot of top mutated genes (>1%) with all patients in substudy with top barplot showing tumour mutation burden. Barplots of top 10 genes showing differences between % mutated in GRs and PRs ESR1[HIGH] (**b**) and mutated in GRs and PRs ESR1[Low] (**c**). Ranking of genes, *p*-values and adjusted *p*-values based on mafCompare function in maftools. Plots include variant classifications (frame-shift deletions−blue, frame-shift insertions−purple, missense−green, nonsense−red, nonstop−light blue, splice site−orange, in-frame deletion−yellow, in-frame insertion−dark red, and multi-hit−black). Source data are provided as a Source data file.

infiltration of immunosuppressive FOXP3+ Tregs and higher expression of negative regulators of T-cell immune functions, such as CTLA4 and PD-L1. In addition, there is a need for prospective trials to better understand this relationship and the relationship between low ER/oestrogen signalling and higher expression of immunosuppressive markers (Figs. 5, 6 and Supplementary Data 2).

Several different immune features indicated a broad involvement of many immune cell types, but they were strongly indicative of involvement of the adaptive immune system. There was a significant correlation between the gene expression for CD3, CD20 and FOXP3 with the mIF analysis, supporting the validity of this multiparameter immunofluorescence. There was also a significant correlation of TILs, T-regulatory cell gene sets, and T-regulatory cell marker FOXP3; however, only significantly higher levels of stromal B-cell marker CD20 were observed in PRs. As expected, these mIF markers were predominantly expressed in the stromal compartment.

A key question is whether the generally increased immune features of the PRs are a determinant of resistance or a marker of it that is correlated with other features that are more directly determining resistance. Somatic mutational burden, neoantigen load, and CNV load have each been shown to weakly correlate with immune infiltration but not specific recurrent genomic alterations (*e.g.* a mutations or site of copy number gain/loss). Here, we have shown a weak but significant association of TILs with *TP53* mutations but there are likely other drivers of immune infiltration. Somatic mutational burden and TILs were not correlated in our dataset, but this analysis is limited by our panel size (87 genes). However, the targeted panel captured regions across the genome at approximately 3 Mbp resolution (Supplementary Data 4) and provided copy number estimates across the genome. We did not observe a correlation of chromosomal instability with TILs nor a difference between chromosomal instability in GRs and PRs. There is a modest but significant difference between GRs and PRs ESR1[HIGH] across several measures of immune infiltration and related

gene expression, but TILs are not likely to be a major driver of Ki67 response when *ESR1* expression is high. It should be noted that overall immune-related features are inversely correlated with oestrogen response (Figs. 3b and 5c). Additional work needs to be undertaken to determine the interaction between expression of oestrogen-responsive genes and immune signalling.

If endocrine resistance is in part driven by aspects of the immune phenotype, such as by the presence of cytokines that impact on BC growth, this would provide both a target for therapy and a factor to be aware of when undertaking clinical trials of immune modulation. It is of particular interest that the IL6/STAT3/JAK3 pathway is strongly associated with the endocrine resistance, as IL6 is known to be secreted by some BC cells in response to oestrogen deprivation[14] and is a stimulant of inflammation[36]. IL6-like cytokines are known to exert their effect through the shared signal transducer IL6ST[37] which was included as one of just four genes in a classifier of endocrine resistance in the neoadjuvant setting[38]. There is a number of clinically registered anti-IL6 and anti-IL6R antibodies that could potentially be exploited in this area; however, caution should be exerted with immune therapies until a better understanding of the biology of these tumours and responses to treatment are known. The results of the randomized phase II SAFIR02-BREAST IMMUNO trial, which randomized patients with HER2- advanced disease to the immune checkpoint inhibitor, durvalumab, or maintenance chemotherapy after a course of induction chemotherapy, suggested a worse progression-free survival in the ER+ group treated with durvalumab.

While our endpoint was the change in Ki67, a number of studies use a high value of Ki67 after 2 weeks, commonly >10%, as indicative of endocrine resistance[39,40]. A high 2-week Ki67 value can be used to identify patients at sufficiently high risk of recurrence on an endocrine agent alone to merit either treatment with an additional agent or different treatment altogether (e.g. chemotherapy)[12,41,42], but it is not indicative of the response to AI. However, these differences in defining

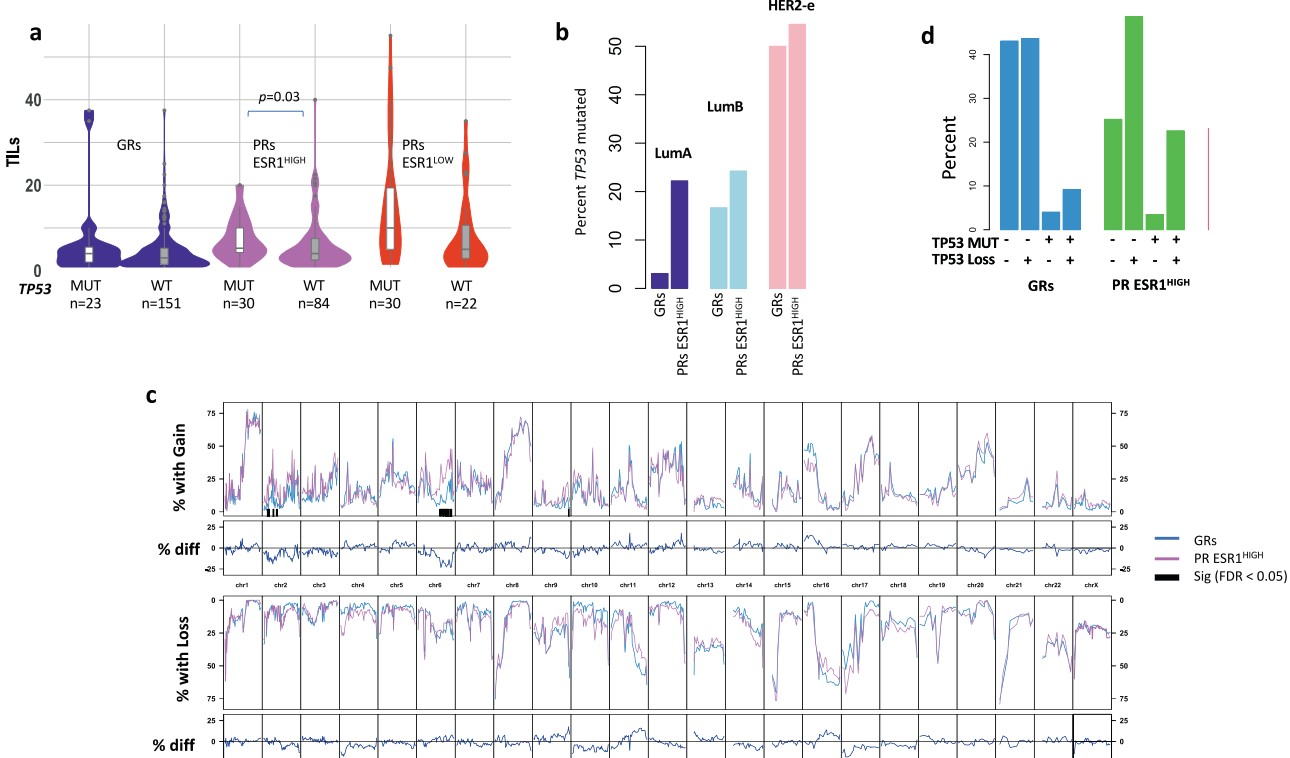

**Fig. 8 | TILs, TP53 mutations and copy number alterations. a** Violin/Boxplot of TILs associated with TP53 status in GRs, PRs ESR1$^{HIGH}$ and PRs ESR1$^{LOW}$. Boxplots present 25th, 50th (median), and 75th percentile values. Whiskers extend no larger than ±1.5 times the inter-quartile range with outliers plotted individually beyond this range. The number of independent samples used for comparisons between GRs and PRs is shown. Significant differences ($p < 0.05$) determined by two-sided Mann–Whitney tests. **b** Barplot of the percent of tumours with *TP53* mutations for LumA, LumB and HER2-E subtypes in GRs, PRs ESR1$^{HIGH}$. **c** Plots of the percent of GRs (blue) or PRs ESR1$^{HIGH}$ (orchid) with gains or losses at individual chromosomal locations. Purple bars highlight regions with significant differences between GRs and PRs ESR1$^{HIGH}$ (Fisher-exact test with FDR/Benjamini and Hochberg adjustment). **d** Percent of patients with *TP53* mutations and/or loss of copy number. Source data are provided as a Source data file.

Ki67 endpoints may not have been of great importance in our study, since less that 1% of GRs in our study had Ki67 > 10% after 2 weeks of AI treatment compared to 90% of PRs.

We found that 1/3 of PRs had low *ESR1* levels and 91% of these were non-Luminal subtypes. Such low levels of the key determinant of response to AIs provide a means of resistance and we therefore excluded these cases from the key comparisons with GRs. This ensured that markers would not be erroneously identified as associated with resistance by virtue of their correlation with low *ESR1* expression. The risk of such spurious associations is emphasized by only 15% of genes that were expressed significantly differently between all PRs and GRs as well as between PRs ESR1$^{HIGH}$ and GRs. This issue should be borne in mind in all studies of endocrine resistance.

It was striking that the proportion of LumA and LumB tumours was not greatly different between GRs and PRs ESR1$^{HIGH}$, although there was an enrichment of HER2-enriched subtypes in PRs ESR1$^{HIGH}$. This is an observation we made previously in a much smaller population[14]. It supports the better outcome of luminal A tumours on endocrine therapy being predominantly due to their better intrinsic prognosis and much less due to a better response to endocrine therapy. Additional work needs to be undertaken across the wider POETIC trial to understand the independent prognostic values of intrinsic subtypes and AI response. As PRs ESR1$^{HIGH}$ showed partial response to AI, it is possible that these tumours may gain more benefit from AI than the non-luminal tumours in PRs ESR1$^{LOW}$.

In addition to the immune hallmarks discussed above, significant associations of PRs ERS1$^{HIGH}$ with G2M checkpoint and mitotic spindle hallmarks are evidence of dysregulated control of proliferation,

despite the balancing of baseline Ki67 levels between the GRs and PRs populations. Patients showing these features might be good candidates for CDK4/6 inhibitors, which we and others have shown to cause profound decreases in Ki67 in tumours showing incomplete suppression with an AI alone.

In the past, many studies of model systems of endocrine resistance have highlighted the prominence of growth factor signalling pathways as causative and attractive for targeting[43–45]. In the current study, we highlighted the greater expression of two growth factor receptors (FGFR4, which is associated with HER2-e subtypes, and EGFR) in PRs ERS1$^{HIGH}$, but these features poorly distinguish between GRs and PRs ERS1$^{HIGH}$ (Fig. 9). Clinical studies in ER + HER2-disease focusing on growth factor receptor pathways have been discouraging.

Our mutational studies did not find any highly significant somatic mutations to be associated with resistance after correcting for multiple analyses when comparing GRs and PRs ESR1$^{HIGH}$, but *TP53* and *RB1* mutations were enriched in PRs ESR1$^{HIGH}$ and *TP53* very significantly enriched in PRs ESR1$^{LOW}$. The significant enrichment of *TP53* mutations in LumA PRs ESR1$^{HIGH}$ suggests that *TP53* may have a major role in response to AI in this subtype and further studies are needed to validate this finding. There was a trend for higher TILs in tumours with *TP53* mutations but only reached significance in PRs ESR1$^{HIGH}$, suggesting *TP53* mutations may play a role in immune infiltration but is unlikely to be a major driver. The low frequency of most mutations is statistically challenging even when comparing subgroups within a large study but can still suggest potential mechanisms to explore in future research.

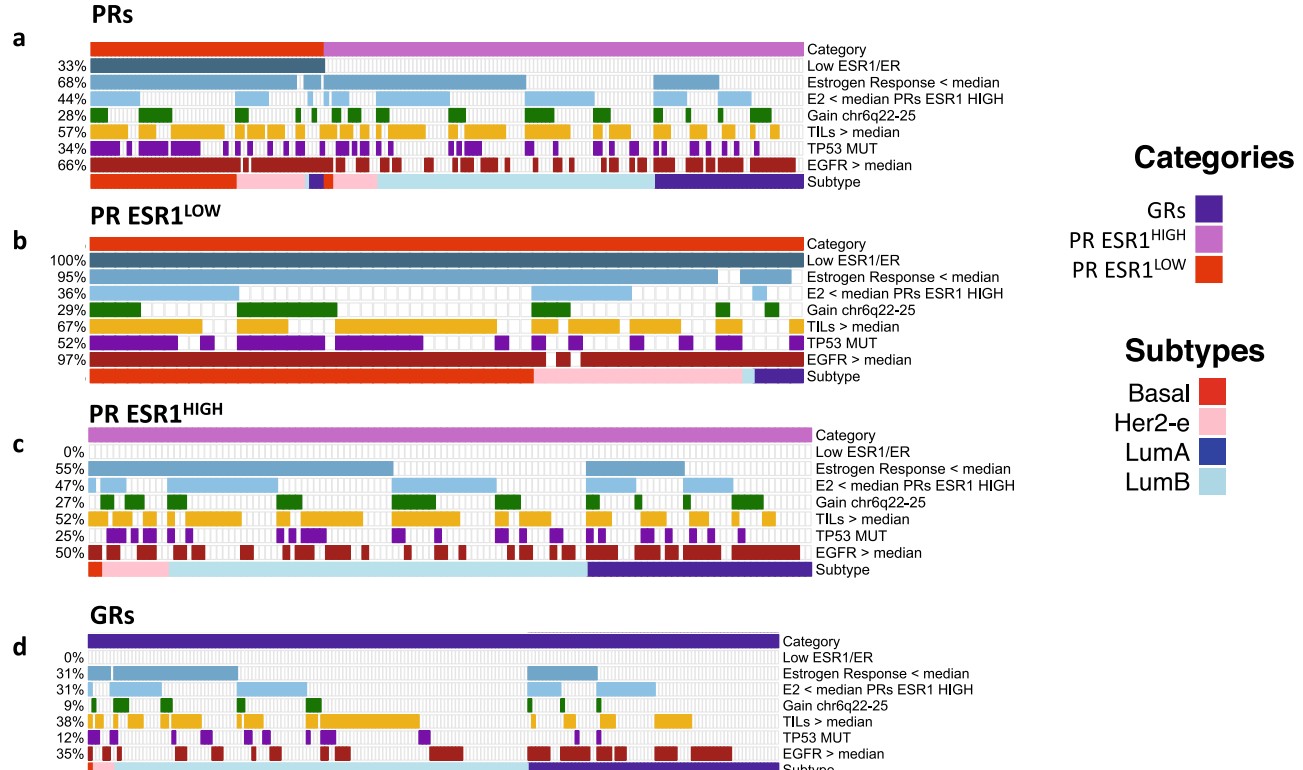

**Fig. 9 | Top features for GRs and PRs categories.** Oncoplot of top features of AI resistance across PRs (**a**), PRs ESR1[LOW] (**b**), PRs ESR1[HIGH] (**c**), and GRs (**d**). Low oestrogen response (based on Hallmark Oestrogen Response Early ssGSEA scores), low E2 levels, high TILs, and high expression of EGFR determined by median expression of these features across all samples. Source data are provided as a Source data file.

We observed that some recurrent copy number alterations occurred at much higher frequencies than somatic mutations in agreement with other studies[28]. Additional studies of regions of 6q are required to understand if these regions play a role in AI resistance or sensitivity. When combining copy number alterations and somatic mutations, there was a significant enrichment of PRs with both TP53 loss and mutations (23% ESR1[HIGH] and 25% ESR1[LOW]) compared to GRs (9%), suggesting p53 status in ER + HER2- can have a major impact on AI response. It has been shown that patients with somatic mutations and loss of wild-type *TP53* have poor outcomes[46], and poor response to AI in these tumours is likely to impact outcomes.

The study presented here is a genomic comparison between GRs and PRs to AI treatment, validates several known factors (low *ESR1*/ER expression, low oestrogen response signalling, higher expression of growth-factor receptors and *TP53* mutations) associated with poor response, and associates additional features with poor response to AI (low oestrogen, immune infiltration, and gains of chr6q) that need further validation. With the emergence of new treatment options for primary ER + HER2- BC, this work is relevant to further treatment of patients and establishing biomarkers that can be developed for patient management. Finally, the work highlights the importance of using the proliferation biomarker Ki67 to determine intrinsic AI resistance, as persistent proliferation after AI treatment is a marker of resistance that is independent of mechanism.

## Methods
Patients provided written informed consent before enrolment and POETIC was approved by the London–South East Research Ethics Committee (reference 08/H1102/37). Samples were selected for analysis from postmenopausal women with primary ER + BC that participated in the POETIC trial (CRUK/07/15; ClinicalTrials.gov, NCT02338310; the European Clinical Trials database, EudraCT2007-

003877-21; and the ISRCTN registry, ISRCTN63882543). The POETIC trial randomized 4480 postmenopausal women with palpable or at least 1.5 cm tumours by ultrasound to receive AI 2 weeks before and 2 weeks after surgery or no perisurgical treatment. See Smith et al.[12] for more details of the full trial. Core-cut biopsies were taken from all patients prior to starting the AI and either a core-cut biopsy or a piece of the excision biopsy was taken for each patient at the time of surgery and was fixed in formalin. Core-cut biopsies at diagnosis and surgery were also placed into RNAlater in a proportion of the patients and have been reported on previously,[47,48] but only Formalin-Fixed Paraffin-Embedded (FFPE) samples were used in the current report. Blood samples were taken prior to starting the AI, immediately prior to surgery and at first follow-up after surgery.

HER2- patients selected for this substudy were taken from the AI treatment arm of POETIC. Also, baseline Ki67 IHC was >10% to minimise imprecision in proportional Ki67 falls. PRs to AI were defined as the patients in the lowest 15% of anti-proliferative response to AI based on the proportional change in Ki67 between baseline and 2 weeks. PRs were matched with GRs within the top 50% of anti-proliferative response based on baseline Ki67 to ensure that the GRs selected were not biased for low proliferation tumours (e.g. Luminal A) (Supplementary Fig. 1b). GRs were matched to PRs based on Ki67 categories (10–20%, 20–30%, and ≥30%) using the ccmatch function in stata.

MIB1 (1:50 dilution) was used as the primary antibody to stain Ki67, and detection was done with the REAL EnVision system, both from DAKO (Glostrup, Denmark until 2016; now Agilent Technologies, Didcot, UK; Dako Cat. M7240). Sample and between-batch quality control scoring was according to methodology on which the International Ki67 in Breast Cancer Working Group Party had based its Ki67 scoring recommendations[12,49]. HER2 status was measured locally using IHC and/or fluorescence in situ hybridization.

 

## RNA and DNA extraction

RNA and DNA were coextracted from three microdissected 10 μm formalin-fixed paraffin-embedded (FFPE) sections, from the baseline and surgery samples of the patients included in the study (Supplementary Fig 1A). The ROCHE High Pure miRNA Isolation kit (Roche, Basel, Switzerland) and the Allprep FFPE kit for DNA (Qiagen) were used following SOP M027 from The Cancer Genome Atlas (TCGA) Program developed by the Biospecimen Core Resource at Nationwide Children's Hospital in Columbus, Ohio. Quantification was done using high-sensitivity RNA and DNA Qubit assays (Thermo Fisher Scientific, Carlsbad, CA) following the manufacturers' protocol.

DNA was extracted from blood using the EZ1 DNA Blood 350 μl kit and EZ1 Advanced XL magnetic bead system (Qiagen) following manufacturer's instructions.

## RNA-seq and analysis

At least 200 ng were used for exon-capture-based RNA-seq library preparation using the SureSelect XT RNA Direct kit and the SureSelect Exome V6 + UTR Capture kit (Agilent, Santa Clara, CA). Libraries were sequenced on the NovaSeq platform (Illumina, San Diego, CA). Salmon was used for quantifying the expression of transcripts[50] based on gencode version 22 GTF transcript annotation. In R (version 4.0.2 using Bioconductor version 3.15), the filterByExpr function in edgeR[51] was used to determine expressed genes and DESeq2[52] was used for detection of differentially expressed genes by the use of negative binomial generalized linear models. DOSE and clusterprofiler[53] were used for gene set enrichment analysis (GSEA) of Hallmark and Gene Ontology gene sets from the Molecular Signature Database[54,55]. Consensus Tumour Microenvironment (TME) was used for generating BC specific generating cancer specific signatures for multiple cell types[25]. Single set GSEA (ssGSEA) enrichment scores,[56] as calculated in the GSVA package in R, were used to represent the degree to which the genes within a gene set were co-ordinately up/down regulated in a sample. Benjamini & Hochberg (1995) method (false discovery rate [FDR]) was used for multiple correction (p-value adjustment).

## Intrinsic subtyping

Intrinsic subtyping of Basal-like (Basal), HER2-enriched (HER2-e), Luminal A (LumA) and Luminal B (LumB) was determined by a variation of gene-level median centering[20] based on 59 samples that were additionally subtyped with the Nanostring BC360 codeset. As our subset of tumours would have different distribution of ER + HER2- subtypes than the TCGA data (due to the relatively high proliferation of our subset) (Supplementary Fig. 2d, e), we identified a small random subset of samples in both the TCGA ER + HER2- and our dataset to use for median centering that would maximise the number of patients with the correct subtype call. A random selection of 5 to 15 samples with BC360 subtypes was taken and matched by subtype distribution to a random selection from a public microarray training set (220arrays_nonUBCcommon+12normal_50g.txt). Subtype calls in the RNA-seq dataset was based on mean centering to the matched microarray data subset. Random selection was done 1.5 million times and the maximum overlap between BC360 and RNA-seq subtype calls (57 out of 59; 97% concordance) was used as the calibration factor to make subtype calls on the whole dataset. The maximum iteration used 8 samples (4 Basal, 3 HER2-enriched and 1 Luminal B), and Pearson correlation was 0.91 for all subtype correlations (Basal, HER2-enriched, Luminal A, and Luminal B) between BC360 and gene-centred RNA-seq calls from the common 59 samples.

## Targeted exome sequencing and analysis

A targeted exome panel was designed covering 87 genes (Supplementary Data 4) selected to include genes with genomic alterations (somatic mutations and/or copy number alterations) associated with de novo resistance to AI. The panel also included genomic regions approximately every 3 Mbp to allow detection of chromosomal instability across the genome. Blood from eight patients was used for controls to detect any systematic biases in the library preparation and sequencing. A total of 250 ng DNA was used for library preparation conducted using the SureSelect XT Low Input Reagent Kit (Agilent, Santa Clara, CA) and target regions of interest captured by Target Enrichment Baits produced by Agilent. Libraries were sequenced on the NovaSeq platform (Illumina, San Diego, CA). Sequencing adaptors were trimmed by trim-galore (www.bioinformatics.babraham.ac.uk/projects/trim_galore/). The BWA software (version 0.7.15) was used to align trimmed fastq sequences to genome version hg38. Protocols based on the GATK version 4.0 to mark read duplicates, recalibrate base quality scores and filter mutation calls were performed[57] on the aligned bam files. Mutect2 from GATK was used to call somatic mutations and Ensembl Variant Effect Predictor[58] was used to annotate the effects of mutations on protein coding genes. Only mutations with 5% allele frequency, observed in at least five alternative reads, high to moderate consequences on a protein sequence, and ExAC and gnomAD allele frequencies below $10^{-5}$ were included for analysis with maftools[59]. Mutations were visually confirmed using the Integrative Genomic Viewer version 2.13.0 (https://software.broadinstitute.org/software/igv/). A panel of normals was generated from the eight blood samples to identify germline mutations and systematic biases. The cnv.hmm method in CNVkit[60] version 0.9.7 was used to call copy number alterations.

## Plasma estradiol

Plasma estradiol levels were measured in baseline samples by liquid chromatography and tandem mass spectrometry in Biochemistry Department in the Wythenshawe Hospital, Manchester University NHS Foundation trust.

**Quantification of estradiol.** Calibrators, quality controls and samples (250 μL) were pipetted into the wells of a 96-well plate and 150 μL of deionized water and 40 μL of 17β-estradiol-2,3,4-$^{13}$C$_3$ working internal standard at a concentration of 10 μg/L were added to each well before the plate was heat-sealed and vortex mixed for 1 min. The contents of each well were then transferred to the corresponding well of a 400 μL supported liquid extraction (SLE) plate. Estradiol was subsequently eluted from the SLE plate using 900 μL methyl tert-butyl ether and collected into a second 96-well plate. The eluate was dried and reconstituted in 75 μL of 40% methanol.

**Chromatography.** An Acuity® ultra-performance liquid chromatography classic system was used for chromatographic separation. Mobile phases consisted of (A) deionized water with 50 μmol/L ammonium fluoride and (B) methanol. The prepared sample (37.5 μL) was injected onto a 2.1 × 150 mm 2.7 μm Waters Cortecs® C18 column coupled to a Waters in-line filter. Starting conditions were 60% mobile phase B at a flow rate of 0.4 mL/min, which was maintained for 3 min before increasing to 95% B. Subsequently, the mobile phase composition was held for 0.7 min before returning to starting conditions to re-equilibrate. The total run time injection-to-injection was 4.2 min.

**Analysis.** Following chromatographic separation, the column eluate was directed into a Waters Acquity® Xevo TQ-XS mass spectrometer. The mass spectrometer was operated in the negative electrospray ionisation mode, the capillary voltage was maintained at 2.5 kV, and the source temperature was 150 °C. The desolvation temperature and gas flow were 650 °C and 1200 L/Hr, respectively. A 1/x weighted linear regression model was applied and the ratio of analyte peak height to internal standard peak height was plotted against estradiol concentration (pmol/L). The limit of quantification was 3 pmol/L.

For the liquid chromatography tandem mass spectrometry assay, the coefficient of variation for estradiol concentrations of 125 pmol/L

was <7% and 10% at 22 pmol/L, and the correlation coefficient ($R^2$) with a commercial immunoassay (Abbott Diagnostics, Maidenhead, UK) was 0.98[61]. The performance characteristics were validated against the FDA's published criteria (https://www.fda.gov/files/drugs/published/Bioanalytical-Method-Validation-Guidance-for-Industry.pdf) and has Medical Laboratory Accreditation (ISO 15189) from the United Kingdom Accreditation Service.

### Tumour-infiltrating lymphocytes

Tumour-infiltrating lymphocytes (TILs) were counted in the stromal compartment (=% stromal TILs) from hematoxylin and eosin (H&E) stained tumour sections following guidelines described by Salgado et al.[26].

### Multiplex immunofluorescence analysis

A multiplex immunofluorescence (mIF) panel consisting of antibodies to ER (6F11), 1.5 µg/ml (Novocastra); CD3 (LN10), 0.3 µg/ml (Novocastra); FOXP3 (236A/E7), 5 µg/ml (Abcam); CD20 (L26), 0.2 µg/ml (Dako); and CD68 (514H12), 0.2 µg/ml (Novocastra), and, respectively, matched Opal pairings of Opal 620, Opal 690, Opal 570, Opal 520 and Opal 780, used as per manufacturer's instructions (Akoya UK) was validated as outlined by Pulswatdi et al. (2020) on whole face, breast resected FFPE sections[62]. Briefly, validation included chromogenic and fluorescent singleplex optimisation, multiplex immunofluorescence optimisation and validation, optimization of high throughput image acquisition and analysis, and testing of the validated workflow, from staining to digital image analysis. Sections were counterstained with DAPI for nuclear detail and mounted in ProLong Gold Antifade Mountant (Thermo Fisher).

Chromogenic immunostained slides were scanned using the Aperio AT2 (Leica Biosystems) at ×20 magnification. All fluorescently labelled slides were scanned on the Vectra Polaris (Akoya) at ×20 magnification, and appropriate exposure times were established by auto-exposing on a tissue known to be positive for all biomarkers. Images were acquired using tile scanning with the 7-colour whole-slide unmixing filters (DAPI + Opal 570/690, Opal 480/620/780, and Opal 520). Spectral unmixing and autofluorescence removal was performed using the synthetic Opal library available in inForm and a tissue-specific autofluorescence image. Resultant image tiles were then stitched together within the open-source software QuPath v0.2.3 to produce a whole-slide multichannel, pyramidal OME-TIFF image[63]. All images were reviewed for quality and consistency before being considered for digital image analysis.

Digital image analysis process took place using QuPath v0.2.3. Briefly, the tissue was annotated as a region of interest (ROI) where areas of necrosis, fat, staining and tissue artefacts, and ductal carcinoma in situ/normal ducts were largely excluded. Following cell identification using the StarDist script, epithelial tumour and stroma ROIs were identified either on epithelial tumour ER pixel value greater than a user defined threshold, or in cases where ER staining was low, using machine learning training of an average five examples of epithelium and five examples of stroma. The resulting images were always visually reviewed for consistency and accuracy. Biomarker positivity was identified using a machine learning classifier, generated using an average of 20 points of positive expression and 20 points of negative/ignore. A composite classifier for FOXP3/CD3 was generated by combining single classifiers and included in the data analysis. Quantification of biomarker output for each case was based on biomarker cell densities (µm²) for tumour or stroma ROIs.

### Statistics and reproducibility

No statistical method was used to predetermine sample size of this substudy of the POETIC trail. RNA and DNA samples were excluded if no tissue was available or nucleotide yields were too low for sequencing, if the library preparation failed or if the read coverage was too low

after sequencing (see Supplementary Fig. 1b for details). Samples were excluded from TILs analysis if an H&E slide was not available ($n = 1$). Estradiol could not be measures in 18 samples and additional samples were excluded ($n = 3$) because an accurate measurement could not be generated. Samples with estradiol >130 pmol/L were also excluded from correlation analysis, as these patients may not be post-menopausal. Seven samples from the mIF analysis did not pass quality control measures and were excluded. Due to the scarcity of the samples, only a single biological replicate could be used in each experiment. Researchers processing RNA/DNA samples, measuring TILs and estradiol, and measuring mIF cell density were blinded to good/poor responder groups.

### Reporting summary

Further information on research design is available in the Nature Portfolio Reporting Summary linked to this article.

## Data availability

The datasets from the POETIC clinical trial are available under restricted access for privacy and legal issues. Access to data can be obtained by submission and approval of a data and sample request form to the POETIC Trial Management Group (TMG), and reasonable academic requests are likely to be approved within a few months, as the TMG meets several times a year. Data files and details to request access are available from the European Genome-Phenome Archive (EGA) which provides a public and permanent archive for sequencing datasets (Full Dataset - EGAD00001010919; RNAseq study - EGAS00001007302; Targeted Exome study - EGAS00001007303). The processed data are provided in the Supplementary Information/Source data files. Source data are provided with this paper.

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

## Acknowledgements

This work was supported by Breast Cancer Now working in partnership with Walk the Walk (E.F.S., E.L.K., K.S., D.E., E.F., A.A., L.Z. and M.D.), Cancer Research UK (M.D., J.M.B., H.T. and M.C.U.C.), Le Cure, the Arthur Foundation (E.F.S.), Breast Cancer Research Foundation and National Institute for Health Research (NIHR) Biomedical Research Centre at The Royal Marsden NHS Foundation Trust and The Institute of Cancer Research, London. The views expressed are those of the authors and not necessarily those of the NIHR or the Department of Health and Social Care. We would also like to acknowledge the RNA-seq and targeted exome sequencing done by the Tumour Profiling Unit at the ICR espe-cially Nik Matthews and Kerry Fenwick, the estradiol assay performed by James Hawley and George Sherly at Manchester University NHS Foun-dation Trust, and the PgR IHC assay performed by Margaret Hills and Simone Detre. We are grateful to the POETIC trial investigators and their support staff and to the POETIC trial patients.

## Author contributions

E.F.S., M.D., I.S., J.R., J.M.B. and M.C.U.C. conceived and supervised the project. H.T. performed selection and matching of samples. E.L.K., K.S., D.E., and E.F. coordinated the acquisition, extraction and distribution of samples. A.A. and L.Z. performed the TILs analysis. P.M. and M.S.T. performed and supervised the mIF analysis. E.F.S. and M.D. wrote the manuscript with support from all co-authors.

## Competing interests

M.D. reports consultancy for Astrazeneca, the ICR Rewards for Inven-tors Scheme for abiraterone. J.M.B. reports grants from Cancer Research UK, during the conduct of the study; grants from Medivation; grants and non-financial support from AstraZeneca, Merck Sharp & Dohme, Puma Biotechnology, Clovis Oncology, Pfizer, Janssen-Cilag, Novartis, and Roche, outside the submitted work. M.S.T. is a scientific advisor to Mindpeak and Sonrai Analytics, and has received honoraria recently from BMS, MSD, Roche, Sanofi and Incyte. He has received grant support from Phillips, Roche, MSD and Akoya. None of these disclosures are related to this work. J.F.R.R. has received consulting fees from, and has performed contracted research on behalf of, AstraZeneca, Bayer, Novartis and Oncimmune; has given expert testi-mony for AstraZeneca; and holds stock with Oncimmune. M.C.U.C. has a patent for Breast Cancer Classifier: US Patent No. 9,631,239 (Method of classifying a breast cancer instrinsic subtype) with royalties paid and receive research funding from NanoString Technologies and veracyte advisory role. M.D., E.F.S. and M.C.U.C. have patent-pending (PCT/EP2021/07368; Treatment response predictive method) for predictive CDK4/6 inhibitor sensitivity assay. All other authors declare no competing interests.
