## [Peer Review File · Nature Communications]

Molecular profiling of aromatase inhibitor sensitive and resistant ER+HER2- postmenopausal breast cancersREVIEWER COMMENTS

Reviewer #1 (Remarks to the Author): expertise in breast cancer subtype analysis

Schuster et al. have evaluated biomarkers for response to neo-adjuvant (NA) aromatase inhibition (AI) in menopausal patients with ER+, HER2- breast cancer. They did so by comparing Ki67-defined good responders with poor responders after 14 days of AI exposure.

The investigated a number of very diverse biomarkers for their ability to discern good responders and poor responders.

This is an important manuscript that deserves publication.

Please find some minor suggestions that would in my opinion add to the value of this manuscript for the clinician-reader.

1. These biomarkers are diverse, from plasma estradiol levels to TILs. They are all biologically sound, but some a priori justification why they were selected (and others not (?), or not reported) seems appropriate.

2. Regarding follow-up and implicit information. The patients data and tumour samples were drawn from patients enrolled in the POETIC study (Lancet Oncology 2020, Smith et al.). This was a large UK-study on the role of NA AI followed by standard adjuvant treatment vs standard adjuvant treatment. This study was negative for its primary endpoint in the sense that there was no long-term advantage of given a short course of NA AI. The study did show that decreasing Ki67 from an initial elevated level identified those patients with an improved outcome as defined by breast cancer recurrence. It is also worthwhile to remind the reader that the follow-up of this study had a median of 62.9 months (or 5 years), implying that, at best, only half of the recurrences have been identified as yet. It is furthermore important that within the same time frame nearly half of the DFS events were not breast cancer related, including 5% of non-breast cancer primaries and 3% deaths. One in 4 patients (26%) in the experimental arm (from which the patients in this study were recruited) also received chemotherapy. As outlined above the study is correct in its title to investigate biomarkers with aromatase inhibitor resistance in the primary tumor. Clearly, this is very relevant as such, but the reader is tempted to related this directly with benefit or lack thereof on risk developing metastasis. The should emphasize the that the benefit of a good response is only shown to predict "early", ie 60 month recurrence

3. In the NA AI treated group with HER2 negative disease the 5 -year recurrence risk was 8.4% for the high-low group (i.e. good responders) and 21.7% in the high-high group (i.e. poor responders). The majority of breast cancer recurrence in the study was due to distant metastasis. In other words the biomarkers associated with a good response to NA AI predicted a benefit on micrometastatic disease. This still leaves some 30% of all recurrences within 5 years unaccounted for. We know from chemotherapy and other drugs (anti-VEGF, anti-HER2, etc ..) how difficult the relation of a biomarker

such as pCR is to DFS or OS in a study population. It seems appropriate to emphasize this. It is not stated in the text but might implicitly be understood in that manner.

4. Is there an imbalance of chemotherapy in both groups in the current analysis, or were pts selected based on the absence of adjuvant chemotherapy ?
5. The Ki67 is mentioned in table 1. Would it be possible to give the range of Ki-67 at base line ? Fig1A. gives this information for the d14 data.
6. The difference in grade 3 cases in the PRs 41% versus 24% in the GR deserves some clarification. In other words this difference in grade was associated/ explained by a difference in Ki-67? Similar remark for the difference in the grade 2 distribution in both groups.
7. The observation of similar increased MKI67 expression and Ki67 IHC in the GR and the PR ESR1 high is an important observation. Is this difference also reflected with similar levels for the progesterone receptor on IHC analysis (cfr Fig 1 E) ?
8. The changes in GEP in the sense of a differential in estrogen response between GR and PR ESR1 high, suggests indeed a disconnect between the ER signaling and proliferation on the PR ESR1 high. Again was there a similar differential response on PgR expression?
9. The intrinsic subtyping results suggest a substantial proportion of HER2-enriched cases in the PR ESR1 high (1% in GR vs 8% in PR ESR1 high). Was there an increased proportion of HER2 1+ and 2+ (IHC) in the PR ESR1, and was there a change in intrinsic subtyping after 2 weeks of AI between the GR and PR ESR1 high? It is perhaps reasonable to cite work on FGFR expression and endocrine resistance in addition to its association with the HER2-e subtype. (Mao et al CCR, 2020; Saez-Cuixé et al, CCR 2022).
10. What is the HOXB13/IL17BR score in both groups ? and the BCI ?
11. The biomarker significance of the increased immune related genes sets and TILs in the PR ESR1 high is important. Others have summarized the role of TILs and prognosis in luminal breast cancer (e.g. Pellegrino et al, 2021). This work adds on its association with endocrine resistance and this is in line with numerous studies combining endocrine treatment with anti-PD1 and/or anti-PD-L1 also in the endocrine adjuvant setting.

Reviewer #2 (Remarks to the Author): expert in breast cancer –omics

Schuster and colleagues analyze features of HR+ breast cancers, selected from a randomized clinical trial, treated with brief pre-operative aromatase inhibitor (AI) therapy. They employ change in Ki67 levels as a surrogate of treatment response and compare a small subset of the tumors which were matched for baseline Ki67 and had either a major decline (GR, n=190) vs. minimal decline (PR, n=177) in

Ki67. The PR tumors were further categorized into those with low ESR1 expression (n=58) and the rest with ESR1 levels comparable to GR tumors. Overall, they found that the PR ESR1 low tumors were distinct from PR ESR1 high tumors and that that latter were generally more similar to GR tumors. PR ESR1 low tumors were far more likely to be basal or HER2 by intrinsic subtype in contrast to the GR and PR ESR1 high which were luminal A/B. A substantial focus of the manuscript is differences between GR and PR ESR1 high tumors. Among other findings, the authors report that compared to GR tumors, PR ESR1 high tumors have a more active immune microenvironment (also seen in PR ESR1 low tumors), lower estradiol levels (restricted to PR ESR1 high but not PR ESR1 low tumors), and p53 mutation with LOH (seen in both PR ESR1 high and ESR1 low tumors).

Major points:

1. The PR ER high and low are lumped together in terms of hormonal therapy response based on Ki67 in this manuscript, but there may be substantial differences between these two groups that are not fully addressed. Is the overall prognosis of these two groups likely to be the same? Is the (presumably low) clinical benefit from hormonal therapy likely to be similar? Would longer AI treatment possibly have demonstrated differences between PR ER high and low tumors? These points should be addressed/discussed.
2. Analyzing the extremes of Ki67 responses seen in a small minority of cases is useful for comparisons but may fail to capture features of the majority of cases. Are most tumors claimed/likely to fall into these 3 categories, or do they represent extremes of a continuum? Putting the analysis in the fuller context of HR+ breast cancer would be important.
3. The differences in TILs seems small in GR vs. PR ESR1 high (11 vs. 16% intermediate) compared to PR ESR1 low (36%). Given a similar lack of Ki67 response in the latter two groups, it does not seem likely that this feature is a major driver of Ki67 response. Can the authors comment?
4. Since only 15 cases from each category are selected for mIF analysis, how can we have confidence that analysis of such a small number of samples is sufficient to provide confirmatory evidence of the molecular analyses as suggested?
5. Unlike the TILs, the p53 genetic profiles seem to be more similar in the non-responsive tumors compared to responsive ones: the number of tumors with both somatic mutations and loss of one or more copy number in TP53 were significantly higher in PRs ESR1 HIGH (23%) and in PRs ESR1LOW (25%) compared to GRs (9%). These data would seem to support p53 status rather than immune profiles as a contributor to lack of Ki67 response. This should be discussed.

Minor Points:

The labeling of the X and Y axis on many of the graphs shown is illegible (far too small in proportion to the graphs), e.g. Fig 1A, 1C; Fig 4A, 4C.

Reviewer #3 (Remarks to the Author): expertise in biostatistical analysis of clinical trials

Schuster et al. presented an impressive study aiming to identify response or resistance biomarkers of aromatase inhibitors (AIs) in ER+ breast cancer. This study provided useful insight into patient response to AI treatment at the molecular, cellular, and functional levels. The results may benefit future clinical trial design, too. Based on a large cohort of good vs. poor responders, this study is one of the largest of its kind. In particular, the authors categorised all PR patients into ESR1 high/low and this is a right move following published evidence on the role of ESR1. Their attention on eliminating the confounding effect from prognostic factors is important to ensure the findings are robust.

While the manuscript is well written in general, the authors need to improve on the following aspects before publication:

- (1) The title used the term “resistance” and in the main text there is a mixed use of “response”. While resistance may refer to both intrinsic and acquired resistance, “response” is probably more accurate and may be used consistently in the manuscript.
- (2) The description of the method to define GR and PR cohorts are different in the Abstract, in the Methods and in the Results.
- (3) Please provide more detail on how to “match” GR and PR based on Ki67. Are they matched on an individual base or a population base?
- (4) Is intrinsic subtyping only carried out on baseline DNA samples only or there are paired samples?
- (5) Need to include more detail on what statistical analysis have been carried out.
- (6) In Fig 1b, I am not sure if MKI67 is differentially expressed from the plot itself. P values were given in page 7 but please provide more detail on what statistical test has been used and whether these p values have been corrected for multiple comparison?
- (7) Instead of Fig 1-D, I recommend also using a PCA plot to illustrate the similarity/difference among the gene profiles of GR/PR ESR1high/PR ESR1low cohort.

(8) The 5th paragraph on page 8 is very hard to understand and needs rewrite. I am interested to see more info on the correlation, i.e., the distribution of correlations rather than the median values alone, to explore whether weak correlations may be explained by patient heterogeneity.

(9) KRAS signalling any discussion?

(10) Please specify the color code used in Figure 3, both in figure and in axis. Clearly, some hallmarks with most of their genes differentially expressed have not been described in the text. Is there a rationale behind?

(11) The legend of Figure 3-D is not correct.

(12) Page 9, paragraph 3, please provide more detail on the statistical test used in Figure 4.

There are plenty of typo mistakes in the manuscript and needs double checking. For example,

(1) Page 3 first sentence of the 2nd paragraph, "the AI treatment group and were HER2-."

(2) sFig 1, "form"

(3) Page 10, last paragraph, "PRs ESR1HIGH (mean 3.7mut/s/tumor) orPRs ESR1HIGH (mean 3.3mut/s/tumor)". One of them should be ESR1low.

Reviewer #4 (Remarks to the Author): technical expertise in mass spectrometry

In the present study authors report immune signatures, low estradiol levels and TP53 mutations in association with AI resistance in primary estrogen receptor-positive HER2-negative postmenopausal breast cancer. The study is conducted to identify features that impact on the response of ER+HER2- to AI, i.e. poor responders vs good responders. The paper is in general well written, topic and results are of high interest.

Baseline levels of estradiol are measured by LC-MS/MS, the gold standard methodology for accurate quantification of estradiol at the very low postmenopausal levels addressed in this study. Although the method is thoroughly described, there are a few minor concerns to be addressed:

1. Limit of quantification (LOQ) is 3 pmol/L (page 5, Analysis), which provides sufficient sensitivity to determine estradiol levels in most postmenopausal samples. How many of the measured samples (if any) were below LOQ? How were results below LOQ treated statistically (Figure 4 A and C)?
2. Precision of the method (CV %) should be reported at relevant levels.
3. Accuracy and traceability of the method should be reported, i.e. with reference to a certified reference standard or an external quality assessment program.

Bjørg Almås, PhD

REVIEWER COMMENTS

Reviewer #1 (Remarks to the Author): expertise in breast cancer subtype analysis

Schuster et al. have evaluated biomarkers for response to neo-adjuvant (NA) aromatase inhibition (AI) in menopausal patients with ER+, HER2- breast cancer. They did so by comparing Ki67-defined good responders with poor responders after 14 days of AI exposure. They investigated a number of very diverse biomarkers for their ability to discern good responders and poor responders.

This is an important manuscript that deserves publication.

Please find some minor suggestions that would in my opinion add to the value of this manuscript for the clinician-reader.

1. These biomarkers are diverse, from plasma estradiol levels to TILs. They are all biologically sound, but some a priori justification why they were selected (and others not (?), or not reported) seems appropriate.

We have added the following text to the introduction to add justification of the biomarkers chosen: (line 101) "To identify correlates of AI response, we undertook analysis of a diverse set of biomarkers, including comprehensive transcriptome RNA sequencing for discovery, plasma estradiol levels due to our previous report of the strong association between estradiol levels and expression of estrogen-responsive genes (Dunbier et al. 2010), immune markers and tumor-infiltrating lymphocytes (TILs) due to our earlier report of an association between immune-related gene expression and AI resistance (Dunbier et al. 2013), and targeted DNA sequencing of genes known to be frequently mutated in BC (Cancer Genome Atlas 2012; Curtis et al. 2012) or have been associated with AI resistance (Ma et al. 2015).

We have also included additional information on the immune cell phenotyping for clarification and added the following text to the mIF Results section: line 297 "Immune phenotyping was based on 5 markers: CD3 which is part of the T-cell receptor complex, CD20 which is a B-cell surface marker, CD68 which is transmembrane glycoprotein that is highly expressed by human monocytes and tissue macrophages, and FOXP3 which is a transcriptional regulator found in immunosuppressive Tregs."

2. Regarding follow-up and implicit information. The patients data and tumour samples were drawn from patients enrolled in the POETIC study (Lancet Oncology 2020, Smith et al.). This was a large UK-study on the role of NA AI followed by standard adjuvant treatment vs standard adjuvant treatment. This study was negative for its primary endpoint in the sense that there was no long-term advantage of given a short course of NA AI. The study did show that decreasing Ki67 from an initial elevated level identified those patients with an improved outcome as defined by breast cancer recurrence. It is also worthwhile to remind the reader that the follow-up of this study had a median of 62.9 months (or 5 years), implying that, at best, only half of the recurrences have been identified as yet. It is furthermore important that within the same time frame nearly half of the DFS events were not breast cancer related, including 5% of non-breast cancer primaries and 3% deaths. One in 4 patients (26%) in the experimental arm (from which the patients in this study were recruited) also received chemotherapy. As outlined above the study is correct in its title to investigate biomarkers with aromatase inhibitor resistance in the primary tumor. Clearly, this is very

relevant as such, but the reader is tempted to relate this directly with benefit or lack thereof on risk developing metastasis. The should emphasize the that the benefit of a good response is only shown to predict “early”, ie 60 month recurrence

To emphasize the benefit of good AI response and put the recurrence timeframe into context, the following text has been added to the Introduction: (line 91) “Ki67 was measured at diagnosis and at surgery in the tumors of these patients and it was confirmed that Ki67 after 2-weeks’ treatment is more prognostic of 5-year recurrence risk than baseline Ki67 in ER+HER2- BC: recurrences were 60% lower for patients with baseline Ki67 $\geq 10\%$ that fell below 10% after 2 weeks of AI (8.4% recurrence within 5 years) compared to patients whose Ki67 remained high ($\geq 10\%$) after AI (21.7% recurrence within 5 years) (Smith, Robertson et al. 2020), suggesting intrinsic AI resistance is a major factor in ‘early’ recurrence (<5 years). Additional follow-up work will be undertaken to determine if the relationship of Ki67 response to AI with later recurrences persists beyond 5 years after diagnosis and to determine associations between clinicopathological/molecular features and recurrence despite good response to AI.”

3. In the NA AI treated group with HER2 negative disease the 5 -year recurrence risk was 8.4% for the high-low group (i.e. good responders) and 21.7% in the high-high group (i.e. poor responders). The majority of breast cancer recurrence in the study was due to distant metastasis. In other words the biomarkers associated with a good response to NA AI predicted a benefit on micrometastatic disease. This still leaves some 30% of all recurrences within 5 years unaccounted for. We know from chemotherapy and other drugs (anti-VEGF, anti-HER2, etc ..) how difficult the relation of a biomarker such as pCR is to DFS or OS in a study population. It seems appropriate to emphasize this. It is not stated in the text but might implicitly be understood in that manner.

Please see response to comment 2. We have added the following text to the introduction to emphasize the importance and limits of good AI response on risk of recurrence within 5 years of diagnosis: (line 97) “Additional follow-up work will be undertaken to determine if the relationship of Ki67 response to AI with later recurrences persists beyond 5 years after diagnosis and to determine associations between clinicopathological/molecular features and recurrence despite good response to AI.”

4. Is there an imbalance of chemotherapy in both groups in the current analysis, or were pts selected based on the absence of adjuvant chemotherapy ?

Selection did not include chemotherapy since that therapy did not relate directly to the change in Ki67 data. However, there is an imbalance of chemotherapy with significantly more chemotherapy in the PRs group. The following text has been added to the results: (line 148) “In general, the GRs and PRs had similar clinicopathological characteristics except for greater number of PRs that were <59 years old, had grade 3 tumors, or had chemotherapy.”

Please also see response to comment #6.

Chemotherapy has been added to Table 1.

	GRs n=190	PRs n=177	
Chemotherapy			
Yes	29% (43)	42% (75)	p=0.0001 (Fisher-exact)
Unknown	2% (3)	1% (2)	

5. The Ki67 is mentioned in table 1. Would it be possible to give the range of Ki-67 at baseline ? Fig1A. gives this information for the d14 data.

Range of Ki67 at baseline and 2wks has been added to Table 1.

	GRs n=190	PRs n=177
Ki67 % Baseline		
Median	26% (range 10% to 73%)	29% (range 10% to 97%)
Ki67 % 2wk		
Median	2% (range 0% to 11%)	24% (range 6% to 95%)

6. The difference in grade 3 cases in the PRs 41% versus 24% in the GR deserves some clarification. In other words this difference in grade was associated/ explained by a difference in Ki-67? Similar remark for the difference in the grade 2 distribution in both groups.

There is a strong association between Ki67 at baseline and tumor grade in the data, and the following text and figure has been added to explain the discrepancy between Ki67 and tumor grade: (line 171) “*MKI67* gene expression and Ki67 IHC were significantly higher in PRs ESR1^{LOW} compared with PRs ESR1^{HIGH} and GRs (FDR=0.003, DESeq2 analysis and $p=5.5 \times 10^{-7}$, Mann-Whitney analysis respectively) but not PRs ESR1^{HIGH} compared to GRs (Fig. 1e,l; Supplemental Table 1). PRs ESR1^{LOW} are associated with a significantly higher percentage of grade 3 tumors compared to GRs ($p=0.0003$, fisher-exact) and PRs ESR1^{HIGH} ($p=0.03$, fisher-exact), and grade 3 tumors are much more likely to be offered chemotherapy.”

7. The observation of similar increased MKI67 expression and Ki67 IHC in the GR and the PR ESR1 high is an important observation. Is this difference also reflected with similar levels for the progesterone receptor on IHC analysis (cfr Fig 1 E)?

The following text has been added for analysis of PgR IHC data and Figure 1 updated to include PgR IHC (Fig 1h): (line 168) “PgR protein expression was also significantly higher in GRs compared with PRs ESR1^{HIGH} and PRs ESR1^{LOW} ($p=1 \times 10^{-5}$ and 3×10^{-23} , respectively; Mann-Whitney test) (Fig. 1h). PgR was not detected using IHC in 2% (4/183) of GRs, 12% (13/111) of PRs ESR1^{HIGH} and 77% (43/56) of PRs ESR1^{LOW} at baseline.”

8. The changes in GEP in the sense of a differential in estrogen response between GR and PR

ESR1 high, suggests indeed a disconnect between the ER signaling and proliferation on the PR ESR1 high. Again was there a similar differential response on PgR expression?

IHC and RNAseq values for PGR were highly correlated for baseline samples (Spearman rho=0.88, pval < 10⁻¹⁰⁰, Supplemental Fig. 2a) suggesting the PgR IHC assay was robust. PgR IHC was generated for a subset of 65 baseline-2wk pairs of samples (30 GRs, 23 PRs ESR1^{HIGH} and 12 PRs ESR1^{LOW}), but was not extended to full set because of the large number of PR tumors (in particular PRs ESR1^{LOW}) that do not express PgR protein at baseline (please see response to comment #7 above) and 2wk timepoint (53% PRs ESR1^{HIGH} and 92% PRs ESR1^{LOW}). We therefore could not calculate change in PgR protein for a significant proportion of the tumors and did not extend the suggested analysis.

9. The intrinsic subtyping results suggest a substantial proportion of HER2-enriched cases in the PR ESR1 high (1% in GR vs 8% in PR ESR1 high). Was there an increased proportion of HER2 1+ and 2+ (IHC) in the PR ESR1, and was there a change in intrinsic subtyping after 2 weeks of AI between the GR and PR ESR1 high? It is perhaps reasonable to cite work on FGFR expression and endocrine resistance in addition to its association with the HER2-e subtype. (Mao et al CCR, 2020; Saez-Cuixé et al, CCR 2022).

For POETIC, HER2 1+ and 2+ IHC data is not available, as HER2 was not processed centrally in our lab for the POETIC trial. However, HER2/ERBB2 gene expression is not significantly higher in PRs ESR1^{HIGH} compared to GRs. A supplemental figure (Supplemental Fig. 3g) and the following text and citations have been added to the text: (line 279) “*FGFR4* has been associated with HER2-e subtypes and AI endocrine therapy resistance (Garcia-Recio, Thennavan et al. 2020, Mao, Cohen et al. 2020, Sanchez-Guixé, Hierro et al. 2022) and is significantly differentially expressed between GRs and PRs ESR1^{HIGH} (FDR=0.0002, DESeq2; Supplemental Table 1). Both *FGFR4* and *CLCA2* (FDR=0.0002 GRs vs. PRs ESR1^{HIGH}) showed high expression specific to HER2-e subtypes but not the *HER2/ERBB2* gene (Supplemental Fig. 3e-g, Supplemental Table 1). Similarly, *FOXC1* expression was specific to Basal subtypes (Supplemental Fig. 3h).”

Intrinsic subtyping of the treated samples is not available at this time, but we will pursue this in future work to confirm the results from previous work showing much higher percentage of LumB subtypes switching to LumA after treatment compared to Her2-enriched subtypes (Bergamino et al. Clin Cancer Res. 2022).

10. What is the HOXB13/IL17BR score in both groups ? and the BCI ?

The following text has been added to the Results section: (line 288) “The *HOXB13/IL17BR* ratio (H/I) has been shown to be predictive of benefit from endocrine therapy and extended endocrine treatment with low scores showing significant benefit (Jerevall et al. 2008). We observed the mean baseline H/I from the RNA-seq data was higher in PRs compared to GRs (mean H/I GRs=0.32, PRs ESR1^{HIGH}=0.39, and PRs ESR1^{LOW}=0.49) with H/I significantly higher in PRs ESR1^{HIGH} and PRs ESR1^{LOW} compared to GRs ($p=0.048$ and $p=0.0006$, respectively; Mann-Whitney), confirming that H/I can predict benefit from endocrine therapy.”

11. The biomarker significance of the increased immune related genes sets and TILs in the PR ESR1 high is important. Others have summarized the role of TILs and prognosis in luminal breast cancer (e.g. Pellegrino et al, 2021). This work adds on its association with endocrine resistance and this is in line with numerous studies combining endocrine treatment with anti-PD1 and/or anti-PD-L1 also in the endocrine adjuvant setting.

The following text has been added to highlight the prognostic value of immune related biomarks: (line 647) “There is growing number of retrospective studies showing that immune-related biomarkers have prognostic value in ER+ BC beyond higher TILs, including poor outcomes associated with higher tumor infiltration of immunosuppressive FOXP3+ Tregs and higher expression of negative regulators of T-cell immune functions, such as CTL4A and PD-L1 (Pellegrino et al, 2021). In addition, there is a need for prospective trials to better understand this relationship and the relationship between low ER/estrogen signaling and higher expression of immunosuppressive markers (Fig. 5,6 and Supplemental Table 2).”

Reviewer #2 (Remarks to the Author): expert in breast cancer –omics

Schuster and colleagues analyze features of HR+ breast cancers, selected from a randomized clinical trial, treated with brief pre-operative aromatase inhibitor (AI) therapy. They employ change in Ki67 levels as a surrogate of treatment response and compare a small subset of the tumors which were matched for baseline Ki67 and had either a major decline (GR, n=190) vs. minimal decline (PR, n=177) in Ki67. The PR tumors were further categorized into those with low ESR1 expression (n=58) and the rest with ESR1 levels comparable to GR tumors. Overall, they found that the PR ESR1 low tumors were distinct from PR ESR1 high tumors and that that latter were generally more similar to GR tumors. PR ESR1 low tumors were far more likely to be basal or HER2 by intrinsic subtype in contrast to the GR and PR ESR1 high which were luminal A/B. A substantial focus of the manuscript is differences between GR and PR ESR1 high tumors. Among other findings, the authors report that compared to GR tumors, PR ESR1 high tumors have a more active immune microenvironment (also seen in PR ESR1 low tumors), lower estradiol levels (restricted to PR ESR1 high but not PR ESR1 low tumors), and p53 mutation with LOH (seen in both PR ESR1 high and ESR1 low tumors).

Major points:

1. The PR ER high and low are lumped together in terms of hormonal therapy response based on Ki67 in this manuscript, but there may be substantial differences between these two groups that are not fully addressed. Is the overall prognosis of these two groups likely to be the same? Is the (presumably low) clinical benefit from hormonal therapy likely to be similar? Would longer AI treatment possibly have demonstrated differences between PR ER high and low tumors? These points should be addressed/discussed.

The following text has been added to the discussion: (line 614)

“While PRs ESR1^{HIGH} and PRs ESR1^{LOW} shared some AI resistance phenotypes at the pathway level (Fig. 3a), the two groups were very distinct on the molecular level

(Fig. 1b) and likely to have different overall prognosis with worse outcome expected for the non-luminal subtypes (Sorlie et al. 2001) that are highly enriched in PRs ESR1^{LOW}.

While GRs clearly benefit from AI treatment, it is not known whether additional decreases in Ki67 might have occurred in PRs ESR1^{HIGH} if longer AI treatment had been given, but in the IMPACT trial AI-induced suppression of Ki67 was only marginally greater after 12/14 weeks than at 2 weeks (Dowsett et al. 2022; Dowsett et al. 2005). Some estrogen response genes (e.g. *PGR*) are still suppressed in PRs ESR1^{HIGH}, suggesting ER signaling is still functional in these tumors but not driving proliferation.”

(line 745) “Additional work needs to be undertaken across the wider POETIC trial to understand the independent prognostic values of intrinsic subtypes and AI response. As PRs ESR1^{HIGH} showed partial response to AI, it is possible that these tumors may gain more benefit from AI than the non-luminal tumors in PRs ESR1^{LOW}.”

2. Analyzing the extremes of Ki67 responses seen in a small minority of cases is useful for comparisons but may fail to capture features of the majority of cases. Are most tumors claimed/likely to fall into these 3 categories, or do they represent extremes of a continuum? Putting the analysis in the fuller context of HR+ breast cancer would be important.

Although a minority of samples were selected our focus was on the 15% of patients that had the poorest response since our experience has been that including the intermediate responders was likely to add noise and possibly mask otherwise clear relationships. We selected 30% of cases from the best 50% of responders rather than selecting the best 15% of responders specifically to avoid selecting extreme responders.

To put the analysis in fuller context, a Supplemental Fig 1a and the following text has been added: (line 142) “In the POETIC trial, 67% of the ER+HER2- BC had Ki67 IHC >10% and were therefore eligible for this study. Thus, the included PRs and GRs represented 10% and 34% of the total ER+HER2- population in POETIC, respectively (Supplemental Fig. 1a).”

3. The differences in TILs seems small in GR vs. PR ESR1 high (11 vs. 16% intermediate) compared to PR ESR1 low (36%). Given a similar lack of Ki67 response in the latter two groups, it does not seem likely that this feature is a major driver of Ki67 response. Can the authors comment?

The following text and additional columns to Fig. 5b-c has been added to the manuscript: (line 472)

“As expected, TILs were highly correlated with T-cells (particularly T-regulatory cells [Tregs]), immune-related Hallmark gene sets, and PI3K/AKT/MTOR signaling, and inversely correlated with estrogen response (Fig. 5b-c).”

Discussion (line 689) “There is a modest but significant difference between GRs and PRs $ESR1^{HIGH}$ across several measures of immune infiltration and related gene expression, but TILs are not likely to be a major driver of Ki67 response when *ESR1* expression is high. It should be noted that overall immune-related features are inversely correlated with estrogen response (Fig. 3b and 5c). Additional work needs to be undertaken to determine the interaction between expression of estrogen responsive genes and immune signaling.”

4. Since only 15 cases from each category are selected for mIF analysis, how can we have confidence that analysis of such a small number of samples is sufficient to provide confirmatory evidence of the molecular analyses as suggested?

It is difficult to have high confidence in such a small number of samples, and more emphasis should be placed on how the mIF data better characterises TILs. Fig 6g has been moved to Supplemental Fig. 7 and the following text has been added to the manuscript to clarify mIF analysis and put this analysis into context: (line 480)

“To better understand and characterize TILs, we randomly selected 25 samples from each of the GR, PRs $ESR1^{HIGH}$ and PRs $ESR1^{LOW}$ populations for mIF analysis and inspected the FFPE blocks for the amount of residual tumor. Those samples with insufficient amount of tumor were excluded and 15 samples were randomly selected from the remainder for each population.

Immune phenotyping was based on 5 markers: CD3 which is part of the T-cell receptor complex, CD20 which is a B-cell surface marker, CD68 which is transmembrane glycoprotein that is highly expressed by human monocytes and tissue macrophages, and FOXP3 which is a transcriptional regulator found in immunosuppressive Tregs. Figure 6A-F illustrates the multiplexed staining of ER, CD3, CD20, CD68, FOXP3 and CD3/FOXP3 co-localization and selection of ER positive regions to separate tumor from stromal compartments. As expected, density of the immunophenotypic features in tumor compartments was lower than in stromal compartments, and density was the highest for CD3 and CD68. There was also a trend for a higher density of immune markers in PRs compared to GRs with significant differences in CD20 stroma density (Supplemental Fig. 7)

Analysis showed a highly significant correlation between TILs and FOXP3 marker density in the stroma compartment (Fig 6g), confirming the association of TILs with Tregs (Fig. 5b).”

5. Unlike the TILs, the p53 genetic profiles seem to be more similar in the non-responsive tumors compared to responsive ones: the number of tumors with both somatic mutations and loss of one or more copy number in TP53 were significantly higher in PRs $ESR1^{HIGH}$ (23%) and in PRs $ESR1^{LOW}$ (25%) compared to GRs (9%). These data would seem to support p53 status rather than immune profiles as a contributor to lack of Ki67 response. This should be discussed.

Please see reply to comment #3 that highlights the lack of strong evidence for immune related features played a major role in response to AI in $ESR1^{HIGH}$ patients. We have also amended the following text to stress the importance of p53 status in AI response

and limits to associations with TILs: (line 764) “Our mutational studies did not find any highly significant somatic mutations to be associated with resistance after correcting for multiple analyses when comparing GRs and PRs ESR1^{HIGH}, but *TP53* and *RB1* mutations were enriched in PRs ESR1^{HIGH} and *TP53* very significantly enriched in PRs ESR1^{LOW}. The significant enrichment of *TP53* mutations in LumA PRs ESR1^{HIGH} suggests that *TP53* may have a major role in response to AI in this subtype and further studies are needed to validate this finding. There was a trend for higher TILs in tumors with *TP53* mutations but only reached significance in PRs ESR1^{HIGH}, suggesting *TP53* mutations may play a role in immune infiltration but is unlikely to be a major driver.”

(line 792): “When combining copy number alterations and somatic mutations, there was a significant enrichment of PRs with both *TP53* loss and mutations (23% ESR1^{HIGH} and 25% ESR1^{LOW}) compared to GRs (9%), suggesting p53 status in ER+HER2- can have a major impact on AI response. It has been shown that patients with somatic mutations and loss of wild-type *TP53* have poor outcomes⁴⁶, and poor response to AI in these tumors is likely to impact outcomes.”

Minor Points:

The labeling of the X and Y axis on many of the graphs shown is illegible (far too small in proportion to the graphs), e.g. Fig 1A, 1C; Fig 4A, 4C.

Labelling adjusted for most figures to increase size of text.

Reviewer #3 (Remarks to the Author): expertise in biostatistical analysis of clinical trials

Schuster et al. presented an impressive study aiming to identify response or resistance biomarkers of aromatase inhibitors (AIs) in ER+ breast cancer. This study provided useful insight into patient response to AI treatment at the molecular, cellular, and functional levels. The results may benefit future clinical trial design, too. Based on a large cohort of good vs. poor responders, this study is one of the largest of its kind. In particular, the authors categorised all PR patients into ESR1 high/low and this is a right move following published evidence on the role of ESR1. Their attention on eliminating the confounding effect from prognostic factors is important to ensure the findings are robust.

While the manuscript is well written in general, the authors need to improve on the following aspects before publication:

(1) The title used the term “resistance” and in the main text there is a mixed use of “response”. While resistance may refer to both intrinsic and acquired resistance, “response” is probably more accurate and may be used consistently in the manuscript.

Text associated with findings from this study has been edited to ensure that it refers to intrinsic/de novo resistance or response to AI. We have also included ‘intrinsic’ resistance to the manuscript title.

(2) The description of the method to define GR and PR cohorts are different in the Abstract, in the Methods and in the Results.

We have amended the description of the GR and PR cohort in the abstract, methods and results to make the descriptions more consistent, although the word count limit in abstract mean that full description cannot be made. The following text is now in the manuscript:

Abstract (line 26) “We compared the 15% of poorest responders (PRs, n=177) as measured by proportional Ki67 changes after 2 weeks of neoadjuvant AI to good responders (GRs, n=190) selected from the top 50% responders in the POETIC trial and matched for baseline Ki67 categories.”

Methods (line 825) “HER2- patients selected for this substudy were taken from the AI treatment arm of POETIC. Also, baseline Ki67 IHC was >10% to minimise imprecision in proportional Ki67 falls. PRs to AI were defined as the patients in the lowest 15% of anti-proliferative response to AI based on the proportional change in Ki67 between baseline and 2 weeks. PRs were matched with GRs within the top 50% of anti-proliferative response based on baseline Ki67 to ensure that the GRs selected were not biased for low proliferation tumors (e.g. Luminal A) (Supplemental Fig. 1). GRs were matched to PRs based on Ki67 categories (10-20%, 20-30%, and ≥30%) using the ccmatch function in stata.”

Results (line 133) “Only AI-treated patients with HER2- tumors, paired baseline and surgery Ki67 available, and baseline Ki67 immunohistochemistry (IHC) >10% (to minimise imprecision in proportional Ki67 falls) were included for selection. The 15% of patients in the AI-treated group that showed the least proportional fall in Ki67 (PRs) were selected and matched to GRs from the 50% of patients showing the greatest proportional fall in Ki67. As the average baseline Ki67 IHC for PRs was more than 50% higher than the average of all baseline Ki67 IHC in POETIC, GRs were matched to PRs based on baseline Ki67 categories (10-20%, 20-30%, and ≥30%) to ensure similar baseline proliferation rates.”

(3) Please provide more detail on how to “match” GR and PR based on Ki67. Are they matched on an individual base or a population base?

The following text has been added to the methods: (line 828) “PRs were matched with GRs within the top 50% of anti-proliferative response based on baseline Ki67 to ensure that the GRs selected were not biased for low proliferation tumors (e.g. Luminal A) (Supplemental Fig. 1). GRs were matched to PRs based on Ki67 categories (10-20%, 20-30%, and ≥30%) using the ccmatch function in stata.”

(4) Is intrinsic subtyping only carried out on baseline DNA samples only or there are paired samples?

Intrinsic subtyping has only been carried out on baseline RNA samples. Work is ongoing to determine intrinsic subtyping in the treated 2wk samples.

(5) Need to include more detail on what statistical analysis have been carried out.

An effort has been made in the text to better specify the statistical analysis used for differential gene expression from RNAseq or for determining the significance of differences of other molecular features between GRs and PRs (e.g. protein expression as determined by IHC or estradiol levels). Please see reply to comment #6 as an example. DESeq2 analysis was applied on RNAseq data to identify differentially expressed genes by use of negative binomial generalized linear models and FDR method for multiple correction. In general, other molecular features did not have a normal distribution and the Mann-Whitney nonparametric test was used to determine the significance of differences between groups. Multiple testing corrections were not applied to the protein (IHC), TILs, mIF or estradiol analysis.

(6) In Fig 1b, I am not sure if MKI67 is differentially expressed from the plot itself. P values were given in page 7 but please provide more detail on what statistical test has been used and whether these p values have been corrected for multiple comparison?

DESeq2 analysis was applied on RNAseq data to identify differentially expressed genes by use of negative binomial generalized linear models and FDR method for multiple correction. MKI67 values were taken from this analysis and the text now better specifies this analysis: (line 171) “*MKI67* gene expression and Ki67 IHC were significantly higher in PRs $ESR1^{LOW}$ compared with PRs $ESR1^{HIGH}$ and GRs (FDR=0.003, DESeq2 analysis and $p=5.5 \times 10^{-7}$, Mann-Whitney analysis respectively) but not PRs $ESR1^{HIGH}$ compared to GRs (Fig. 1e,i; Supplemental Table 1).”

(7) Instead of Fig 1-D, I recommend also using a PCA plot to illustrate the similarity/difference among the gene profiles of GR/PR $ESR1^{high}$ /PR $ESR1^{low}$ cohort.

Fig. 1D has been moved to Supplemental Fig. 2 and replaced with a PCA plot (now Fig 1J) and the following text has been added: (line 203) “In addition, principal component analysis (PCA) shows clear separation between PRs $ESR1^{LOW}$ and tumors expressing high *ESR1* (GRs and PRs $ESR1^{HIGH}$) (Fig. 1j).”

(8) The 5th paragraph on page 8 is very hard to understand and needs rewrite. I am interested to see more info on the correlation, i.e., the distribution of correlations rather than the median values alone, to explore whether weak correlations may be explained by patient heterogeneity.

Please see reply to comment #10 below.

(9) KRAS signaling any discussion?

Please see reply to comment #10 below.

(10) Please specify the color code used in Figure 3, both in figure and in axis. Clearly, some

hallmarks with most of their genes differentially expressed have not been described in the text. Is there a rationale behind?

Figure 3 has been updated and colors of labels now reflect the Hallmark categories (proliferation, signaling, immune, etc.). The following text has been amended to clarify why only some hallmarks were discussed: (line 309) “Of particular note, about half of the hallmarks that were upregulated hallmarks in PRs ESR1^{HIGH} compared with the GRs were associated with immune processes, and these immune hallmarks were highly significantly correlated with each other (FDR<0.05). In addition, the immune hallmarks were correlated with genes upregulated after KRAS activation (KRAS Signaling Up) and genes regulated by NF-κB in response to tumor necrosis factor alpha (TNFA) (TNFA Signaling via NF-κB). Both KRAS Signaling Up and TNFA Signaling via NF-κB hallmarks were also significantly higher in PRs ESR1^{HIGH}, and the genes within these hallmarks significantly overlapped with immune hallmarks, including interferon gamma response (FDR=2×10⁻⁷ and FDR=3×10⁻³⁰, respectively; hypergeometric distribution calculation). Similarly, hypoxia is higher in PRs ESR1^{HIGH}, consistent with our earlier observation of a hypoxia metagene being correlated with poor Ki67 suppression by AI (Ghazoui et al. 2011), and positively correlated with immune hallmarks, as are two other hallmarks (apical junction and epithelial-mesenchymal transition) that showed correlation with hypoxia. Xenobiotic metabolism was higher in PRs ESR1^{HIGH} and showed only modest correlation with immune associated hallmarks. In contrast, estrogen response hallmarks were inversely correlated with several immune-related gene sets (Fig. 3b). Finally, the two gene signatures associated with proliferation (G2M checkpoint and mitotic spindle) showed higher expression in PRs ESR1^{HIGH} and were strongly correlated with one another but showed little correlation with any of the other signatures.

Correlations were also calculated between the hallmarks that were significantly different in the comparisons between any of the Ki67 response groups using data from all GRs and PRs samples. In general, the correlations are similar to those found in Figure 3b but with stronger negative correlations being observed between estrogen response and immune hallmarks (Supplemental Figure 4a). The heatmap in Supplemental Fig. 4b showed the distribution of correlations and heterogeneity between patients and those patients that have high expression of interferon response genes but medium to low expression of Interleukin-2/ Interleukin-6 (IL2/IL6) signaling genes.”

(11) The legend of Figure 3-D is not correct.

The legend for Fig 3D has been corrected and now states “Figure 3. **Hallmark GSEA and TME analysis.** **a** Plot of normalized enrichment score (NES) of Molecular Signature Database (MSD) Hallmark gene sets with significance of enrichment (adjusted-p value) for all PRs vs GRs, GRs vs PRs ESR1^{LOW} and GRs vs PRs ESR1^{HIGH} and PRs ESR1^{LOW} vs PRs ESR1^{HIGH}. Hallmarks with adjusted p < 0.25 in any comparison shown. Hallmark genes sets with adjusted p-values < 0.05 are shown (red to blue coloring based on significance; grey if > 0.05). Hallmarks are colored by biological process category. **b** Plot of single samples GSEA score correlations between MSD Hallmark gene sets for GRs and PRs ESR1^{HIGH}. Plot of log2 FC PRs ESR1^{HIGH} – GRs for individual genes within MSD Hallmark gene sets that are significantly different between PRs ESR1^{HIGH} and GRs in GSEA **(c)** and all

ConsensusTME BC gene sets (d). FDR values from GSEA analysis. Hallmarks are colored by biological process category.”

(12) Page 9, paragraph 3, please provide more detail on the statistical test used in Figure 4.

The following text has been added to the main text and figure legend: Results (line 403) “Figure 4a shows there was a significantly higher mean level of estradiol in the GRs than in the PRs $ESR1^{HIGH}$ ($p=0.003$, Mann-Whitney test), but no difference of GRs from the PRs $ESR1^{LOW}$ in which signaling through ER would be considered minimal. However, unlike the comparison of *ESR1* expression, there was no distinct cut-off below which GRs were very unlikely to be represented. Estradiol levels were significantly correlated with expression of estrogen response early (FDR=0.01, Spearman) (Fig. 4b) and late (FDR=0.02, Spearman) GSEA Hallmarks but no other hallmarks in tumors with high *ESR1* expression. Therefore, low expression of estrogen response genes may be in part due to the effect of the lower plasma estradiol levels in the PRs $ESR1^{HIGH}$ group. In addition, two out of the top three genes that correlated with plasma estradiol levels in GRs and PRs $ESR1^{HIGH}$ are known to be regulated by estrogen and ER (*PGR* $\rho=0.26$; *GREB1* $\rho=0.27$, Spearman) and are also expressed in significantly lower levels in PRs $ESR1^{HIGH}$ compared to GRs (Supplemental Table 2). There is a trend for lower estradiol in PRs $ESR1^{HIGH}$ compared to GRs regardless of subtype and a significant difference in LumB tumors ($p=0.004$, Mann-Whitney) (Fig. 4c).”

There are plenty of typo mistakes in the manuscript and needs double checking. For example, (1) Page 3 first sentence of the 2nd paragraph, “the AI treatment group and were HER2-.”

The manuscript has also received additional checks to ensure typos are removed.

Text in Methods section now states (line 825) “HER2- patients selected for this substudy were taken from the AI treatment arm of POETIC.”

(2) sFig 1, “form”

Text now states “from”.

(3) Page 10, last paragraph, “PRs $ESR1^{HIGH}$ (mean 3.7muts/tumor) or PRs $ESR1^{HIGH}$ (mean 3.3muts/tumor)”. One of them should be $ESR1^{LOW}$.

Text now states (line 547) “There was a significant difference ($p=0.02$, Mann-Whitney) between the number of mutations in GRs (mean 2.7 mutations/tumor) compared to PRs (mean 3.6 mutations/tumor), but not between GRs and PRs $ESR1^{HIGH}$ (mean 3.7 mutations/tumor) or PRs $ESR1^{LOW}$ (mean 3.3 mutations/tumor).”

Reviewer #4 (Remarks to the Author): technical expertise in mass spectrometry

In the present study authors report immune signatures, low estradiol levels and TP53 mutations in association with AI resistance in primary estrogen receptor-positive HER2-negative postmenopausal breast cancer. The study is conducted to identify features that

impact on the response of ER+HER2- to AI, i.e. poor responders vs good responders. The paper is in general well written, topic and results are of high interest.

Baseline levels of estradiol are measured by LC-MS/MS, the gold standard methodology for accurate quantification of estradiol at the very low postmenopausal levels addressed in this study. Although the method is thoroughly described, there are a few minor concerns to be addressed:

1. Limit of quantification (LOQ) is 3 pmol/L (page 5, Analysis), which provides sufficient sensitivity to determine estradiol levels in most postmenopausal samples. How many of the measured samples (if any) were below LOQ? How were results below LOQ treated statistically (Figure 4 A and C)?

Only two samples were below LOQ (3pmol/L) and these were set to 3pmol/L in Fig. 4a and c.

2. Precision of the method (CV %) should be reported at relevant levels.

The following text has been added to the Methods section: (line 1013) “For the liquid chromatography tandem mass spectrometry assay, the coefficient of variation for estradiol concentrations of 125 pmol/L was <7% and 10% at 22 pmol/L,…”

3. Accuracy and traceability of the method should be reported, i.e. with reference to a certified reference standard or an external quality assessment program.

The following text has been added to the Methods section: (line 1014) “… and the correlation coefficient (R^2) with a commercial immunoassay (Abbott Diagnostics, Maidenhead, UK) was 0.98 (Owen, Wu, and Keevil 2014). The performance characteristics were validated against the FDA’s published criteria (<https://www.fda.gov/files/drugs/published/Bioanalytical-Method-Validation-Guidance-for-Industry.pdf>) and has Medical Laboratory Accreditation (ISO 15189) from the United Kingdom Accreditation Service.”

REVIEWERS' COMMENTS

Reviewer #1 (Remarks to the Author):

All questions have addressed appropriately.

Manuscript is acceptable for publication.

Reviewer #2 (Remarks to the Author):

The manuscript has been substantially clarified and improved in response to the critiques. Overall, the critiques have been satisfactorily addressed and the paper addresses an important topic.

Reviewer #3 (Remarks to the Author):

I am pleased to find all my concerns appropriately addressed and the manuscript satisfactory to be published in its current form.

Reviewer #4 (Remarks to the Author):

Our original concerns have been addressed in the revision, and we have no further comments.